# A computationally efficient method for probabilistic local warming projections constrained by history matching and pattern scaling, demonstrated by WASP-LGRTC-1.0

Philip Goodwin[1], Martin Leduc[2], Antti-Ilari Partanen[3], H. Damon Matthews[4] and Alex Rogers[5]

[1]School of Ocean and Earth Science, University of Southampton, Southampton, SO14 3ZH, UK
[2]Ouranos, Montreal, Canada
[3]Climate System Research, Finnish Meteorological Institute, Helsinki, Finland
[4]Department of Geography, Planning and Environment, Concordia University, Montreal, Canada
[5]Department of Computer Science, University of Oxford, Oxford, UK

*Correspondence to*: Philip Goodwin (p.a.goodwin@soton.ac.uk)

**Abstract.** Climate projections are made using a hierarchy of models of different complexities and computational efficiencies. While the most complex climate models contain the most detailed representations of many physical processes within the climate system, both parameter space exploration and Integrated Assessment Modelling require the increased computational efficiency of reduced-complexity models. This study presents a computationally efficient method for generating probabilistic projections of local warming across the globe, using a pattern scaling approach derived from the Climate Model Intercomparison Project phase 5 (CMIP5) ensemble, that can be coupled to any efficient model ensemble simulation of global mean surface warming. While the method can project local warming for arbitrary future scenarios, using it for scenarios with peak global mean warming $\leq$ 2°C is problematic due to the large uncertainties involved. First, global mean warming is projected using a $10^3$-member ensemble of history-matched simulations with an example reduced complexity Earth system model: the Warming Acidification and Sea-level Projector (WASP). The ensemble-projection of global mean warming from this WASP ensemble is then converted into local warming projections using a pattern scaling analysis from the CMIP5 archive, considering both the mean and uncertainty of the Local to Global Ratio of Temperature Change (LGRTC) spatial patterns from the CMIP5 ensemble for high-end and mitigated scenarios. The LGRTC spatial pattern is assessed for scenario dependence in the CMIP5 ensemble using RCP2.6, RCP4.5 and RCP8.5, and spatial domains are identified where the pattern scaling is useful across a variety of arbitrary scenarios. The computational efficiency of our WASP/LGRTC model approach makes it ideal for future incorporation into an Integrated Assessment Model framework, or efficient assessment of multiple scenarios. We utilise an emergent relationship between warming and future cumulative carbon emitted in our simulations to present an approximation tool making local warming projections from total future carbon emitted.

## 1 Introduction

The dominant climate projections, used by the 5th Assessment Report (AR5) of the Intergovernmental Panel on Climate Change (IPCC, 2013), are made using the Climate Model Inter-comparison Project phase 5 (CMIP5) ensemble (Taylor et al, 2012). However, due to their high level of complexity, state-of-the-art CMIP5 Earth System Models (ESMs) are computationally demanding, and thus cannot be used on a regular basis to inform decision makers about the impacts of arbitrary carbon-emission scenarios.

While a couple of years separate the different generations of CMIP-like experiments, many applications rather require climate simulations to be generated within a much shorter time frame. For instance, impact assessments may require climate projections for scenarios not considered by the CMIP5 experiments, for example scenarios designed to meet Paris Climate Agreement targets and maintain global mean surface warming below 1.5 or 2 °C (e.g. van Vuuren et al., 2018; Brown et al., 2018; Nicholls et al., 2018; Goodwin et al., 2018a), and physical climate simulations are required within Integrated Assessment

Models exploring the coupled economic, societal, ecological and climate systems (e.g. van Vuuren et al., 2018; van Vuuren et al., 2017; McJeon et al., 2014).

To generate computationally efficient climate simulations, a range of lower-complexity – but numerically more efficient –
climate models have been developed. They generally use a reduced spatial resolution and/ora simplified representation of processes included within the complex models (e.g. Smith, 2012; Meinshausen et al., 2011a; Goodwin et al., 2018b).

For example, the highly efficient MAGICC6 climate model uses an upwelling-diffusion representation of the ocean and an hemispherical averaged spatial resolution (Meinshausen et al., 2011a). MAGICC6 has been configured to emulate an ensemble
of the more complex Climate Model Intercomparison Project phase 3 (CMIP3) climate models (Meinshausen et al., 2011a; 2011b), but at a fraction of the computational expense. To generate spatial projections using MAGICC, a pattern scaling approach (e.g. Herger et al., 2015) is applied to emulate the spatial climate patterns from the CMIP3 models (e.g. Fordham et al. 2012): the regional climate SCENarioGENerator (SCENGEN). This MAGICC6 (and combined MAGICC6/SCENGEN) climate model is computationally efficient enough to usefully couple into Integrated Assessment Model (IAM) frameworks,
including the IMAGE and MESSAGE frameworks (van Vuuren et al., 2017; McJeon et al., 2014). A key goal of IAMs is to explore consequences of the coupled human-climate system, through coupling representations of the physical climate system with the biosphere and human/society interactions, often including energy generation and land-use changes.

A recent study (Goodwin et al., 2018b) takes a different approach to making future projections of global mean surface warming,
using the computationally efficient Warming Acidification and Sea-level Projector (WASP) climate model (Goodwin, 2016; Goodwin et al., 2017). In Goodwin et al. (2018b) the efficient WASP model is configured, not by tuning the parameters to emulate existing complex climate models (e.g. Meinshausen et al., 2011a; 2011b), but instead by history matching (Williamson et al., 2015) the efficient model to real world data. Goodwin et al. (2018b) first generate one hundred million ($10^8$) simulations using WASP, by varying the model properties with a Monte Carlo approach. This includes an input distribution for climate
sensitivity drawn from geological evidence (PALEOSENS, 2012). These $10^8$ simulations are then integrated from year 1765 to 2017, and each of them is checked against a set of historic observational reconstructions of surface warming (Hansen et al., 2010; Smith et al., 2008; Vose et al., 2012), ocean heat uptake (Levitus et al., 2012; Giese et al., 2011; Balmaseda et al., 2013; Good et al., 2013; Smith et al., 2015; Cheng et al., 2017) and carbon fluxes (IPCC, 2013; le Quéré et al., 2016). Only those WASP simulations that are consistent with the observational constraints are extracted to form the final history-matched
ensemble of around $3\times10^4$ simulations (Goodwin et al., 2018b, see Supplementary Table 3 therein). This final history matched ensemble is then used to make future projections (Goodwin et al., 2018b). Note that the WASP ensemble is not configured to emulate the performance of more complex models, but to be consistent with observations of the real climate system.

The WASP model (Goodwin, 2016) produces projections for global mean surface warming only (Goodwin et al., 2018b), so
to gain information to calculate local warming we here apply a pattern scaling tool. Leduc et al (2016) have recently shown that the spatial pattern of warming across CMIP5 models is relatively robust even though the average warming varies widely between ensemble members. Using the well-known pattern scaling approach (Tebaldi and Arblaster, 2014), Leduc et al. (2016) calculated the spatial pattern of the Local to Global Ratio of Temperature Change (LGRTC) that represented the CMIP5 ensemble, including both the mean and standard deviation in this spatial pattern.


Globally, the near-linear sensitivity of mean surface warming to cumulative carbon emissions is expressed via the Transient Climate Response to cumulative CO2 Emissions (TCRE in °C per 1000PgC), which is estimated to be in the range 0.8 to 2.5 °C per 1000PgC (IPCC, 2013; Matthews et al, 2009). One approach to generating local warming projections from carbon

emission scenarios is to simply multiply the LGRTC characteristic of the CMIP5 ensemble (Leduc et al, 2016) by the estimated range for the TCRE and by the cumulative carbon emissions. However, this approach cannot be used to investigate or simulate several phenomena of potential interest. Firstly, the effective TCRE depends on the ratio of $CO_2$ to non-$CO_2$ radiative forcing (Williams et al. 2017a). Therefore, while the efficient climate models can be applied to investigate future warming for arbitrary

scenarios, the TCRE cannot be applied unless it is for a scenario for which the TCRE is already estimated (e.g. Matthews et al. 2009; Williams et al., 2017a), for example the defined Representative Concentration Pathway (RCP) scenarios (Meinshausen et al. 2011c) or an idealised scenario with 1% per year increase in $CO_2$ concentration (1pctCO2; Taylor et al, 2012) and no other forcing. Secondly, studies indicate that there can be a period of continued surface warming following cessation of annual carbon emissions (Frölicher et al., 2014; Williams et al., 2017b). This phenomenon cannot be explored

using the TCRE alone, but is represented within efficient climate models such as WASP (Williams et al., 2017b). Thirdly, there is evidence that in some circumstances there is a path-dependence of surface warming from cumulative emissions (Zickfield et al, 2012), for example where cooling following negative emissions may not re-tracethe previous warming pathway. Again, this phenomenon is not captured within a constant TCRE framework, but may be explored with climate models. Thus a TCRE framework is applicable for certain situations, including idealised scenarios where the TRCE has already

been established, but in the general case a time-dependent Earth system model is required.

In this study, we present a new method for combining the LGRTC approach with an arbitrary efficient Earth system model to generate computationally efficient local warming projections for arbitrary forcing scenarios. Using the WASP model as our example efficient Earth system model, the combined WASP/LGRTC model makes local warming projections that are history

matched to constrain the global mean surface warming (Goodwin et al., 2018b) and pattern scaled to the CMIP5 ensemble to generate the local information (Leduc et al., 2016). Our efficient method of ensemble generation is able to produce warming-projections to year 2100 for arbitrary future carbon-emission scenarios in a matter of seconds on a standard desktop computer (with the computational efficiency of the particular, WASP, efficient model used). An approximation tool is also presented making local warming projections based on future cumulative carbon emitted, for idealised scenarios where the TCRE has

been pre-established.

Section 2 describes the spatial warming patterns analysed for RCP4.5 (Thomson et al., 2011) and RCP8.5 (Riahi et al., 2011) scenarios in 22 CMIP5 models, following the methodology of Leduc et al. (2016). Section 3 describes our methods for producing an ensemble of warming projections for any locality using the combined WASP/LGRTC Earth system model, while

Section 4 presents the approximation approach for cases when the TCRE is pre-established. Section 5 discusses the wider implications of this study.

## 2. Spatial warming patterns in the CMIP5 ensemble for RCP2.6, RCP4.5 and RCP8.5

Leduc et al (2016) demonstrated the utility of considering the spatial warming over time as a product of the global mean

warming, $\Delta \overline{T}(t)$, and the spatial pattern of the Local to Global Ratio of Temperature Change, LGRTC($x,y$), in the CMIP5 ensemble,

$$\Delta T(x, y, t) = \Delta \overline{T}(t) \times \text{LGRTC}(x, y). \tag{1}$$

The mean and standard deviation in LGRTC were analysed across 12 CMIP5 models (Leduc et al, 2016), under a 1 per cent per year increase in atmospheric $CO_2$ concentration (1pctCO2; Taylor et al, 2012). To first order, for scenarios that do not reach peak warming before 2100, the mean LGRTC can be treated as being independent of time and emission scenarios (Leduc et al, 2016, 2015).

Here, the spatial warming patterns in 22 CMIP5 models (see Supplementary Table S1) are examined for RCP4.5 (Thomson et al., 2011) and RCP8.5 (Riahi et al., 2011) scenarios that contain also non-CO2 forcings from for example anthropogenic non-CO2 greenhouse gas and aerosol emissions. We evaluated the LGRTC comparing mean global temperature between years 2006-2025 and 2079-2098. RCP2.6 data was not available for models CESM1-BGC, inmcm4, and IPSL-CM5B-LR. For the other 19 models, we calculated the RCP2.6 LGRTC for the temperature peak period, defined as a 20-year time window with the maximum time-average global mean surface air temperature. Different models had the peak temperature at different times so the we identified the peak individually for each model run. For most models, the peak in 20-year running-mean global temperature was around year 2070. For MIROC-ESM, CSIRO-Mk3-6-0, CCSM4, MRI-CGCM3, and CSIRO-Mk3-6-0 the period with the highest mean temperature was the years 2079-2098. The same reference period (2006-2025) was used as with the calculation of LGRTC using the end-of-the-century period for RCPs 4.5 and 8.5. Note that for RCP2.6 the LGRTC was calculated using the peak temperature period, rather than 2079-2098, because the 2078-2098 period was a similar temperature, or colder, than 2006-2025 in some models, making the calculation of LGRTC impractical since the denominator of the calculation (the global mean temperature change) was too small or negative.

Figure 1 shows the multi-model mean LGRTC ($\mu_{LGRTC}$) and multi-model standard deviation in LGRTC ($\sigma_{LGRTC}$) for the RCP4.5, RCP8.5 and RCP2.6 scenarios. With exception of oceanic regions where non-linear processes have important impacts on the climate sensitivity, such as the sea-ice albedo feedback in the Arctic and the meridional overturning circulation in the north Atlantic (Leduc et al., 2016), LGRTC is very similar in the RCP4.5 and RCP8.5 scenarios (Fig. 1, b,c). The uncertainty of the warming patterns within each scenario, defined as standard deviation of LGRTC within the model ensemble ($\sigma_{LGRTC}$), was largest in the Arctic Ocean and in the Southern Ocean for RCP4.5 and RCP8.5 (Fig 1e,f). The spatial average of the multi-model standard deviation was larger in the RCP4.5 than in RCP8.5 over most areas of the globe. Over continents, it was around 0.15-0.45 in RCP4.5 and mostly below 0.3 in RCP8.5. The RCP2.6 scenario shows greater multi-model mean LGRTC at low latitudes (Fig. 1a,b,c), and has more inter-model variation in the LGRTC at high latitudes (Fig. 1, d,e,f), compared to the RCP4.5 and RCP8.5 scenarios.

The difference in LGRTC between two scenarios, relative to the multi-model variation within a scenario, is expressed via a spatially averaged ratio of $\left|\mu_{LGRTC,i}(x,y) - \mu_{LGRTC,j}(x,y)\right|/\sigma_{LGRTC,i}(x,y)$, where $i$ signifies the reference scenario and $j$ the scenario for comparison. Table 1 expresses how many multi-model standard deviations each of the three scenarios multi-model mean LGRTC lies relative to the other scenarios. Considering the mid-range scenario (RCP4.5) as the reference, the LGRTC for RCP8.5 lies a spatial average of just 0.17 standard deviations away from RCP4.5 (Table 1), indicating that the variation in LGRTC between models within the RCP4.5 scenario is more significant than the variation between RCP4.5 and RCP8.5 scenarios. In contrast, the LGRTC for the RCP2.6 scenario lies 2.8 standard deviations away from RCP4.5 (Table 1). The multi-model-mean LGRTC for RCP4.5 and RCP8.5 scenarios lie a spatial average of 0.78 and 0.75 standard deviations away from the RCP2.6 scenario respectively (Table 1). Note that the asymmetry in Table 1, with lower difference when RCP2.6 is used as the reference scenario, reflects the larger values of $\sigma_{LGRTC}$ in the RCP2.6 scenario (Fig. 1d,e,f).

**3 Local warming projections in the pattern-scaled WASP/LGRTC ensemble**

The aim here is to generate computationally efficient future projections of local warming across the globe, including a measure of the uncertainty in those local warming projections. This is distinct from generating a spatial warming projection that is internally physically consistent, maintaining physically plausible teleconnections between warming at different locations. Each CMIP5 model simulation creates a unique internally physically consistent spatial warming pattern for the prescribed forcing. When projecting local warming, including a measure of uncertainty, one method is to use information on the average and

variation in the LGRTC information from multiple CMIP5 models (Figs. 1, 2). However, as soon as the information from multiple CMIP5 models are combined, the averaged result may not be internally physically consistent in terms of the spatial pattern of warming.

Section 3.1 describes how an observation-constrained projection of global mean surface warming is generated, including uncertainty. Section 3.2 then combines this global mean projection with the LGRTC information from the CMIP5 models (Section 2, above) to generate local warming projections.

### 3.1 Generating global mean warming projections

The WASP Earth system model comprises an 8-box representation of carbon and heat fluxes between the atmosphere, ocean and terrestrial systems (Goodwin, 2016), with surface warming solved via a functional equation linking warming to cumulative carbon emitted (Goodwin et al, 2015). For the terrestrial system, carbon uptake by photosynthesis is dependent on temperature and $CO_2$, while carbon release via respiration is temperature dependent. Heat and carbon initially enters the ocean at the surface ocean mixed layer. Once in the surface ocean mixed layer, heat and carbon are exchanged with the sub-surface ocean

regions over e-folding timescales that vary between each simulation in the ensemble.

Here, the WASP model configuration of Goodwin et al. (2018b) is used. First, WASP is used to generate $3 \times 10^6$ initial simulations in a Monte Carlo approach, each one integrated from years 1765 to 2017. A history matching approach (Williamson et al., 2015) is then adopted to assess these initial $3 \times 10^6$ simulations for observational consistency against historic

warming, ocean heat uptake and carbon fluxes (Supplementary Table S2; and see Goodwin et al., 2018b for how the history matching approach is applied to the WASP model). A total of $1 \times 10^3$ simulations are found to be observationally consistent, such that their simulated values of surface warming, ocean heat uptake and carbon fluxes are consistent within observational uncertainty (Supplementary Table S2; Goodwin et al., 2018b).

The $1 \times 10^3$ observation-consistent simulations are extracted to form the final history matched ensemble. This ensemble is then integrated into the future to generate the distribution of global mean surface warming over time,  (Figure 3). The distributions of global mean surface warming, $\Delta \overline{T_i}(t)$, projected by this configuration and history matching approach using the WASP ensemble, are similar to the CMIP5 projectionsfrom highly complex ESMs for the four RCP scenarios (Goodwin et al., 2018b, see figure 2 therein). However, possibly because the WASP projections are more tightly constrained to observations, they

show reduced ensemble spread in future warming compared to the CMIP5 ensemble.

### 3.2 Generating local warming projections

We now utilise projected distributions from the same configuration of the WASP model to calculate distributions of local warming across the globe using the LGRTC pattern scaling approach of Leduc et al (2016). The aim is to generate an ensemble

of projections of local warming at time $t$ for some scenario, $\Delta T_i(x, y, t)$, by using the history matched WASP projections of $\Delta \overline{T_i}(t)$, and the mean and standard deviation of the LGRTC for the CMIP5 models, $\mu_{LGRTC}(x,y)$ and $\sigma_{LGRTC}(x,y)$ respectively (Figs 2-3).

#### 3.2.1 Constructing the LGRTC suitable for a range of non-RCP scenarios

The aim here is to apply a LGRTC calculation that will likely apply for multiple potential future scenarios, not just the three RCP scenario evaluated (Figure 1). To achieve this, we now combine the LGRTC fields for the different RCP scenarios to find aggregated LGRTC fields, considering the spatial domain over which this is likely to be feasible. The mean and standard

deviations for the LGRTC at location *x,y*, in the new combined scenarios are calculated from the underlying RCP scenarios, using

$$\mu_{LGRTC}(x,y) = \sum_{i=1}^{n} \mu_i(x,y)/n \tag{2}$$

and

$$\sigma_{LGRTC}(x,y) = \sqrt{\sum_{i=1}^{n}\left(\sigma_i(x,y)\right)^2} \tag{3}$$

where *n* is the number of underlying RCP scenarios used.

The domain of the LGRTC in the new combined scenarios is assumed valid where the variation in LGRTC between underlying RCP scenarios is less than the variation ascribed within the new scenario, $\sigma_{LGRTC}(x,y)$. This is calculated such that $\mu_{LGRTC}(x,y)$ exists where the variation between the mean of the LGRTC from the different scenarios is less than the combined standard deviation in the LGRTC $|\mu_j - \mu_k|/\sigma_{LGRTC} < 1.0$, for all combinations of two underlying RCP scenarios *j* and *k*.

This method (eqs. 2 and 3) is used to generate LGRTC fields for three potential generic scenarios (Figure 2). First, a scenario for any arbitrary future warming scenario (*arbitrary* scenario) is constructed by combining all three RCP scenarios (RCP2.6, RCP4.5 and RCP8.5) (Fig. 2a, d, g). Second, a LGRTC scenario for warming consistent with Paris Climate Agreement targets of 1.5 and 2 °C (*generic ≤ 2°C* scenario) is constructed by combining RCP2.6 and RCP4.5 (Fig. 2, b,e,h), the two RCP scenarios containing (at least some) model simulations that do comply with the Paris Agreement. Lastly, a LGRTC scenario for future warming that is likely to exceed the Paris Climate Agreement targets (*generic ≥ 2°C* scenario) is constructed using RCP4.5 and RCP8.5 (Fig. 2, c,f,i), the scenarios where most (RCP4.5) or all (RCP8.5) simulations exceed 2°C.

The *arbitrary* and *generic ≤2°C* LGRTC scenarios are problematic to use in practice. Firstly, the large values of $\sigma_{LGRTC}(x,y)$ across many regions, especially over land (Fig. 2d,e), make any local warming projection highly uncertain. The high $\sigma_{LGRTC}(x,y)$ values arise from the high inter-model variation in the LGRTC in the RCP2.6 scenario (Fig. 1b, eqs. 2,3). Secondly, both *arbitrary* and *≤2°C* generic scenarios have regions that fail the validity criteria, $|\mu_j - \mu_k|/\sigma_{LGRTC} < 1.0$, and so are outside of the prescribed LGRTC domains (Fig. 2a,b, white regions). The largest of these regions lie in the low latitude oceans, with most areas outside the valid domain being marine. Most densely populated areas on land are within the valid domain, and so the LGRTC approach can be applied to project future local warming. Areas outside the applicable domain (Fig. 2a,b) are generally where inter-model variation, $\sigma_{LGRTC}(x,y)$, is small (Fig. 2d,e and Fig. 1d,e,f), rather than where inter RCP scenario variation, $\mu_j - \mu_k$, is large (Fig.1, a,b,c).

The *generic ≥2°C* LGRTC pattern, a combination of RCP4.5 and RCP8.5 (eqs. 2,3) is usable in practice for more generic future warming scenarios. The *generic ≥2°C* LGRTC pattern retains a small $\sigma_{LGRTC}(x,y)$ (Fig. 2 compare f to d,e) and, due to the similarities between LGRTC fields for RCP4.5 and RCP8.5 scenarios (Fig. 1, Table 1), the LGRTC pattern for the generic ≥2°C scenario remains within the validity criteria for the entire globe (Fig. 2c,f,i). The *generic ≥2°C* LGRTC pattern (Fig. 2) assumes idealised future pathways within the range of the RCP4.5 and RCP8.5 scenarios (Figure 3b,c), including a similar ratio of $CO_2$ to non-CO2 radiative forcing and spatial emissions of anthropogenic aerosols. This *generic ≥2°C* LGRTC field should not be used for extreme scenarios that differ widely from the underlying societal assumptions of the RCP sceanrios, for example in their spatial aerosol forcing (e.g. see Liu et al., 2018).

### 3.2.2 Combining the LGRTC patterns with a probabilistic ensemble for global mean warming

Here, we combine LGRTC patterns (Figs. 1, 2) with global mean warming projections from an efficient Earth system model. While we use the WASP model here, other efficient models could be used. For the $i$th ensemble member of this history matched WASP ensemble, the WASP/LGRTC projection of local warming at location $x, y$, $\Delta T_i(x, y, t)$, is constructed using both the mean and standard deviation in the LGRTC from the CMIP5 models,

$$\Delta T_i(x, y, t) = \Delta \overline{T_i}(t) \times [\mu_{LGRTC}(x, y) + z_i \sigma_{LGRTC}(x, y)] , \tag{4}$$

where $z_i$ is randomly chosen from a standard normal distribution. This distribution of local warming at time $t$, (eq. 4), includes both the uncertainty in global mean warming in the WASP ensemble (Figure 3; Goodwin et al., 2018b), and uncertainty in the spatial pattern of warming, $\sigma_{LGRTC}$, which is statistically derived from the CMIP5 ensemble (Figure 2; Leduc et al, 2016). Note that eq. (2) does not assume that the distribution of global mean temperature projections, $\Delta \overline{T_i}(t)$, from the efficient Earth system model is Gaussian. The distribution of $\Delta \overline{T_i}(t)$ may not be Gaussian if, for example, the assumed climate sensitivity distribution has a long tail of high values (e.g. see Knutti et al., 2017). Thus, this method for generating the local warming distribution, eq. (2), can be applied to any arbitrary distribution of global mean surface warming from any arbitrary efficient climate model. If, however, the distribution of global mean surface temperature, $\Delta \overline{T_i}(t)$, were known in advance to be Gaussian, then it may be preferable to generate the local warming distribution, $\Delta T_i(x, y, t)$, by multiplying the Gaussian distributions for global warming and LGRTC directly, rather than applying eq. (2) which multiplies the individual values within each distribution.

The full WASP/LGRTC-ensemble local warming projections for RCP 4.5 and RCP 8.5 are given in Fig. 4, which shows the mean, 17th and 83rd percentile of the warming across the globe from the $1 \times 10^3$ WASP/LGRTC ensemble members. To generate the local projections (eq. 4) for RCP4.5 and RCP8.5, we apply the pattern scaling analysed from the CMIP5 models for the appropriate scenario (Fig. 2). In both scenarios, there is more uncertainty, that is a higher range of responses between the 17th and 83th percentiles, in local warming at high northern latitudes (Fig. 4), consistent with this area showing a larger ensemble spread between CMIP5 models (Fig. 1).

The radiative forcing from aerosols can be highly localised, and so the ensemble mean and variation of local warming, $\mu_{LGRTC}(x,y)$ and $\sigma_{LGRTC}(x,y)$ in eq. (4), depends on how the $CO_2$ and non-$CO_2$ agents evolve in the scenario. For that reason, we include local warming patterns for the 1pctCO2 scenario as well as the RCP4.5, RCP8.5 and *generic ≥2°C* scenarios in the pattern scaling for the WASP/LGRTC model code (https://doi.org/10.5281/zenodo.4001523) This allows future users to choose the spatial pattern scaling that is most suitable for their scenario. The next section utilises the *generic ≥2°C* LGRTC pattern (Fig. 2c) to project spatial warming patterns for scenarios where the cumulative carbon emission is specified.

### 4. Approximation for arbitrary cumulative carbon emission scenarios

This section explores further increasing the computational efficiency for making spatial warming projections for idealised future scenarios, by approximating to the history matched WASP ensemble projections of global mean surface warming as function of cumulative carbon emitted after 2018, $I_{em}$ in PgC.

The distribution of global mean surface warming in the WASP/LGRTC ensemble is approximately normally distributed for the RCP scenarios (Figure 3a). The history matched ensemble mean and standard deviation, $\mu_{\Delta \overline{T}}$ and $\sigma_{\Delta \overline{T}}$ respectively, are

both well approximated by second order polynomials in cumulative carbon emitted (Figure 3b,c). The ensemble mean warming projections is given by,

$$\mu_{\Delta\overline{T}}(I_{em}) = a_1 I_{em}^2 + b_1 I_{em} + c_1 \,, \qquad (5)$$

and the ensemble standard deviation by,

$$\sigma_{\Delta\overline{T}}(I_{em}) = a_2 I_{em}^2 + b_2 I_{em} + c_2 \,, \qquad (6)$$

where $a_1$=3.50257×10⁻⁷, $b_1$=2.50924×10⁻³, $c_1$= 1.02159, $a_2$= 2.14129×10⁻⁸, $b_2$=2.28077×10⁻⁴ and $c_2$=8.79361×10⁻² for RCP8.5. Both the RCP4.5 and RCP2.6 scenarios see very similar warming per unit future carbon emitted to RCP8.5, while the RCP6.0 scenario sees only slightly less warming per unit future carbon emitted (Figure 3b,c).

Therefore, for emission scenarios over the 21st century in which the ratio of radiative forcing from sources other than CO2 to cumulative carbon emitted during the 21st century lies within the range of the RCP scenarios, the distribution of global mean surface warming from the history matched WASP ensemble can be approximated by quadratics in future carbon emitted (eqs. 5 and 6; Fig. 3)

The mean warming at location $x,y$ is calculated by simply multiplying the mean of the $1\times10^3$ WASP ensemble members of the global average warming by the CMIP5 mean of the LGRTC,

$$\mu_{\Delta T}(x, y, I_{em}) = \mu_{\Delta\overline{T}}(I_{em}) \times \mu_{\mathrm{LGRTC}}(x, y) \,. \qquad (7)$$

The standard deviation in local warming at location $x,y$ after cumulative emissions $I_{em}$, $\sigma_{\Delta T}(x, y, I_{em})$, is then calculated from the standard deviation in the global average warming in the $i$ ensemble members, $\sigma_{\Delta\overline{T}}(I_{em})$, and the standard deviation in the LGRTC, $\sigma_{LGRTC}(x,y)$, using,

$$\sigma_{\Delta T}(x, y, I_{em}) = \mu_{\Delta T}(x, y, I_{em}) \sqrt{\left(\frac{\sigma_{\Delta\overline{T}}(I_{em})}{\mu_{\Delta\overline{T}}(I_{em})}\right)^2 + \left(\frac{\sigma_{\mathrm{LGRTC}}(x,y)}{\mu_{\mathrm{LGRTC}}(x,y)}\right)^2} \,. \qquad (8)$$

Note that in this approximation tool the uncertainty in local warming is calculated directly by multiplying the assumed Gaussian distributions of LGRTC and global mean warming, eq. (8). This is unlike the uncertainty calculation for the generic method, eq. (4), which does not assume a Gaussian distribution for global mean warming. Applying equations (7) and (8) provides a method to approximate local warming projections as a function of the future carbon emitted after the start of 2018 (Figure 5a; code available at https://doi.org/10.5281/zenodo.4001523)), including uncertainty in the warming at any location (Figure 5b). This method assumes idealised future pathways within the ranges of the RCP4.5 and RCP8.5 scenarios (Figure 3b,c), including a similar ratio of CO2 to non-CO2 radiative forcing. The *generic ≥2°C* scenario LGRTC field (Fig. 2) is applied (Fig. 4), and as such the approximation tool should be utilised for cumulative carbon emission values that give a best estimate for global mean warming of 2°C or more. While this approximation tool (Fig. 5; eqs. 5-8) is not as general as the full WASP/LGRTC Earth system model in its potential applications, we anticipate it will still be a useful tool for back-of-the-envelope approximations and pedagogical applications.

## 5. Discussion

A highly computationally efficient Earth System Model has been presented for projecting local warming projections, based on a history matched global mean warming projection using an efficient ESM (Goodwin, 2016; Goodwin et al., 2018b) and pattern scaling of the CMIP5 ensemble (Leduc et al., 2016): the WASP/LGRTC model. Along with the full WASP/LGRTC model is an easy to use normal error propagation approximation variant producing projected ranges of both global mean warming and the spatial distribution of warming for future cumulative carbon-emission values.

The WASP/LGRTC model presented here is an alternative to existing efficient climate models. For example, the MAGICC6/SCENGEN efficient model is often configured as an 'emulator' of the CMIP3 ensemble (Meinshausen et al, 2001a,b): the MAGICC6/SCENGEN model parameters are tuned such that the ensemble members emulate the properties of the more complex CMIP3 models in both global mean warming and spatial warming patterns. However, even the most complex of climate model ensembles show discrepancy to observations (Goodwin et al, 2018b), and this discrepancy will be reproduced by an emulating ensemble. In contrast, the WASP/LGRTC model is not tuned to emulate more complex models. Instead the WASP model parameters are empirically constrained using the observed histories of warming, heat uptake and carbon fluxes to generate global mean surface warming projections (Goodwin et al, 2018b). Meanwhile, the LGRTC spatial pattern applies the mean and standard deviation in the spatial warming from across the CMIP5 ensemble (Leduc et al, 2016), but does not seek to emulate any specific CMIP5 model within any specific WASP/LGRTC ensemble member.

At present, the WASP model requires prescribed radiative forcing from greenhouse gasses and agents other than $CO_2$, for example methane or aerosols (Goodwin, 2016; Goodwin et al, 2018b). Future work will seek to implement an emission-based representation of other significant greenhouse gases and aerosols, allowing the WASP/LGRTC model to explore a wider range of future scenarios.

Both the WASP/LGRTC model and the quadratic approximation to WASP/LGRTC model are easy to use. The full WASP/LGRTC model can quickly generate output for arbitrary future scenarios, while the approximated model makes projections for different future cumulative emissions assuming that the relative $CO_2$ and non-$CO_2$ radiative forcing is in the range of the RCP8.5, RCP4.5 or RCP2.6 scenarios (Figure 3b,c compare black dashed line to red, orange and purple).

We anticipate that our full and approximated models will be beneficial both for scientific and pedagogical applications, where available computational resources or climate-model expertise exclude the use of highly complex models

**Code availability.** Versions of the WASP model is available from the public GitHub repository at https://github.com/WASP-ESM/WASP_Earth_System_Model. The specific code for both the WASP/LGRTC combined model approach used in this study, and the local warming projection approximation tool, are archived on Zenodo (https://doi.org/10.5281/zenodo.4001523).

**Author Contributions.** PG conducted the numerical modelling and coded the WASP/LGRTC model and approximation tool, with input from AR. AIP, MD and HDM analysed the spatial patterns in the CMIP5 models, and supplied the spatial arrays used by the WASP/LGRTC model and approximation tool. All authors contributed to writing the manuscript.

**Competing interests.** Authors declare no competing interests

**Acknowledgements.** PG acknowledges funding from UK NERC grant NE/N009789/1 and combined UK Government Department of BEIS and UK NERC grant NE/P01495X/1. ML thanks Ouranos and Concordia University. AIP was supported by Emil Aaltonen Foundation, The Fonds de recherche du Quebec - Nature et technologies (grant number: 200414), Concordia Institute for Water, Energy and Sustainable Systems (CIWESS), and Academy of Finland (grant number: 308365). We acknowledge the World Climate Research Programme's Working Group on Coupled Modelling, which is responsible for CMIP, and we thank the climate modelling groups (listed in Supplementary Table S1) for producing and making available their model output. For CMIP the U.S. Department of Energy's Program for Climate Model Diagnosis and Intercomparison provides coordinating support and led development of software infrastructure in partnership with the Global Organization for Earth System Science Portals.

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

| Reference Scenario | RCP2.6 | RCP4.5 | RCP8.5 |
|---|---|---|---|
| RCP2.6 | - | 0.78 | 0.75 |
| RCP4.5 | 2.83 | - | 0.17 |
| RCP8.5 | 2.15 | 0.41 | - |

**Table 1: The difference between one scenario LGRTC and another, expressed as the spatially averaged number of multi-model standard deviations in LGRTC the multi-model mean LGRTC is from the second scenario relative to the first: $\int \left| \frac{\mu_j - \mu_i}{\sigma_i} \right| dA / \int dA$, where $A$ is surface area, $\mu_j$ and $\mu_i$ are the mean LGRTC of scenarios $i$ and $j$, and $\sigma_i$ is the standard deviation in LGRTC for scenario $i$.**

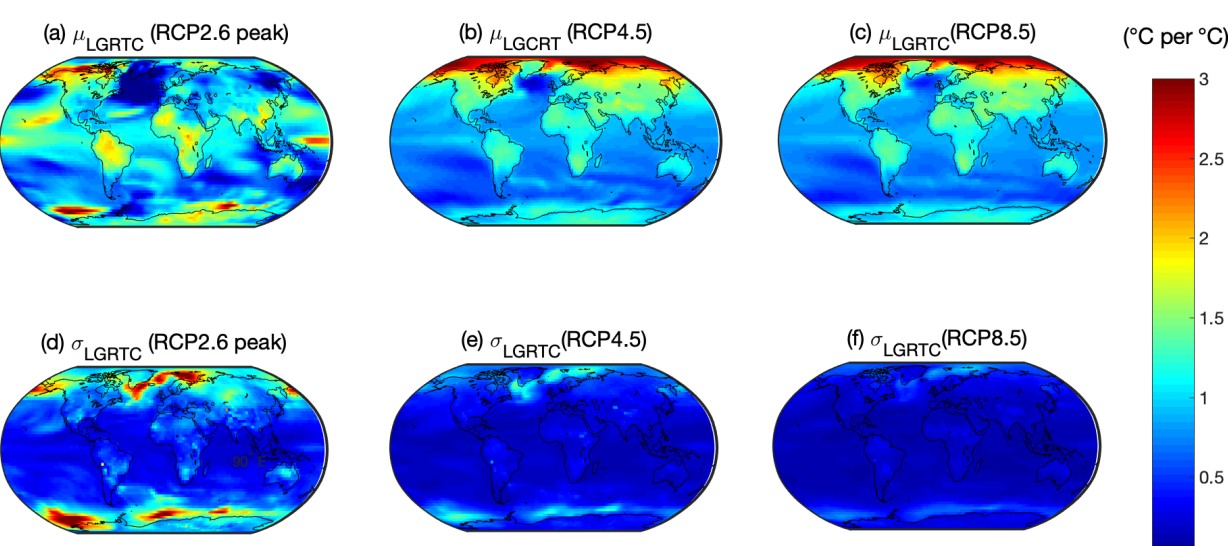

**Figure 1: The LGRTC in RCP2.6, RCP4.5 and RCP8.5 scenarios analysed from a multi-model ensemble of CMIP5 simulations. (a), (b) and (c) show the multi-model mean LGRTC, $\mu_{LGRTC}$, while (d), (e) and (f) show the multi-model standard deviation in LGRTC, $\mu_{LGRTC}$, for each scenario.**

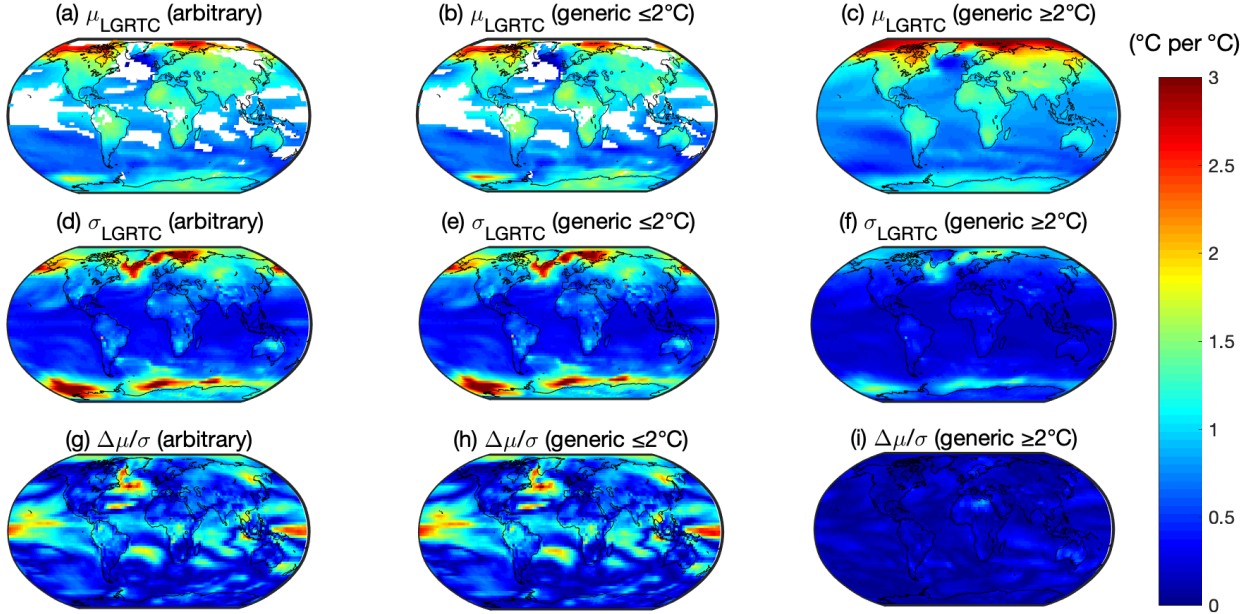

**Figure 2: The LGRTC in the *arbitrary*, *generic ≤2°C* and *generic ≥2°C* scenarios. Panels (a), (b) and (c) show the scenario mean LGRTC. Panels (c), (d) and (e) show the scenario standard deviation in LGRTC. Panels (g), (h) and (i) show the ratio of the maximum absolute discrepancy in the mean LGRTC from the underlying RCP scenarios, $\Delta\mu$, to the standard deviation in the LGRTC, $\sigma$, in the combined scenario: $\Delta\mu/\sigma$.**

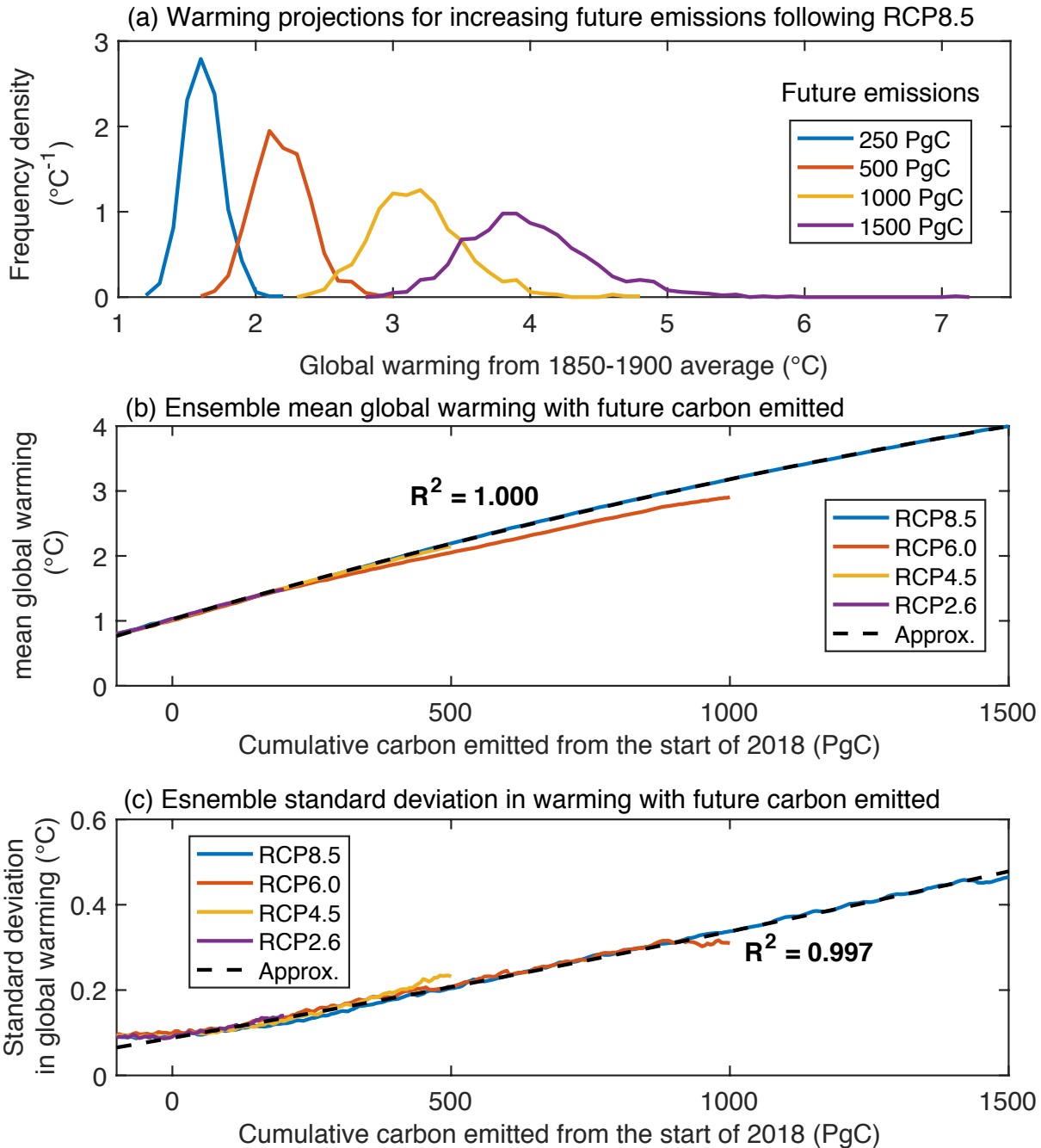

**Figure 3: Projections of global mean surface warming from the history matched WASP ensemble for different future carbon emission sizes. (a) Frequency distributions of projected warming in the WASP ensemble for different future carbon emission sizes after the start of 2018. (b) Ensemble-mean global warming as future cumulative carbon emitted increases. (c) Ensemble standard deviation in global warming as future carbon emitted increases. (b) and (c) show the RCP8.5 (blue), RCP6.0 (red), RCP4.5 (orange) and RCP2.6 (purple) scenarios. A quadratic approximation, eq. 3 for (b) and eq. 4 for (c), is a good fit to the RCP8.5 scenario (thin black line). All panels show warming calculated relative to the 1850-1900 average.**

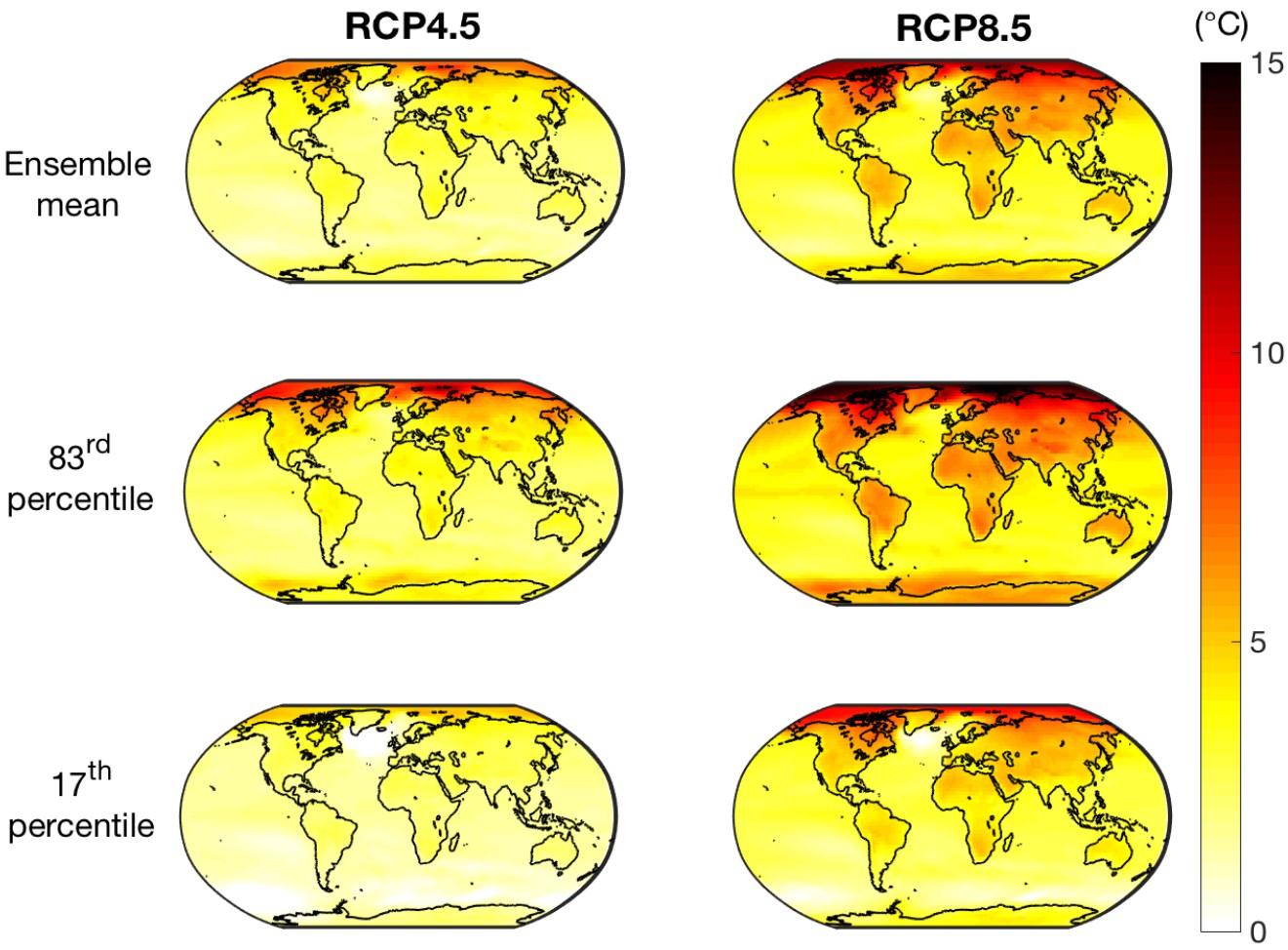

**Figure 4: Projected warming for the period 2081-2100 relative to the 1850-1900 average from $1\times10^3$ history matched simulations of the ultra-fast WASP/LGRTC ensemble. The left-hand column is for the RCP4.5 scenario and the right-hand column is for the RCP8.5 scenario. The top, middle and bottom rows represent the mean, 83rd percentile and 17th percentile of the model ensemble.**

## (a) Central warming projection, future carbon emission = 600PgC

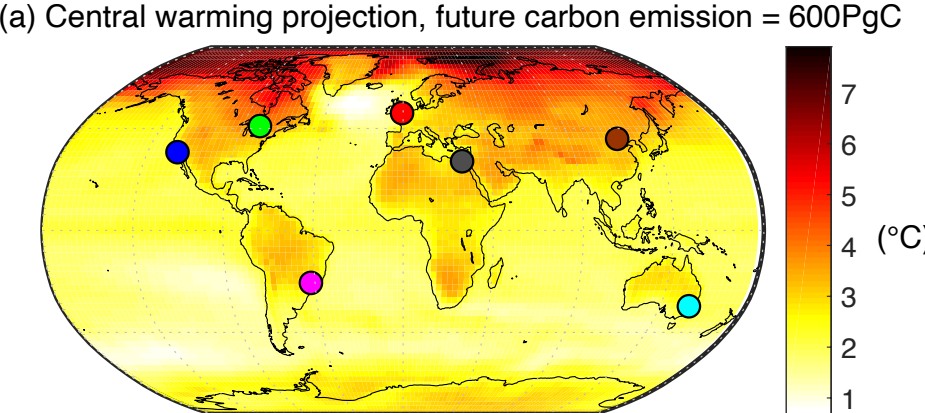

## (b) Probabilistic warming projection, future carbon emission = 600PgC

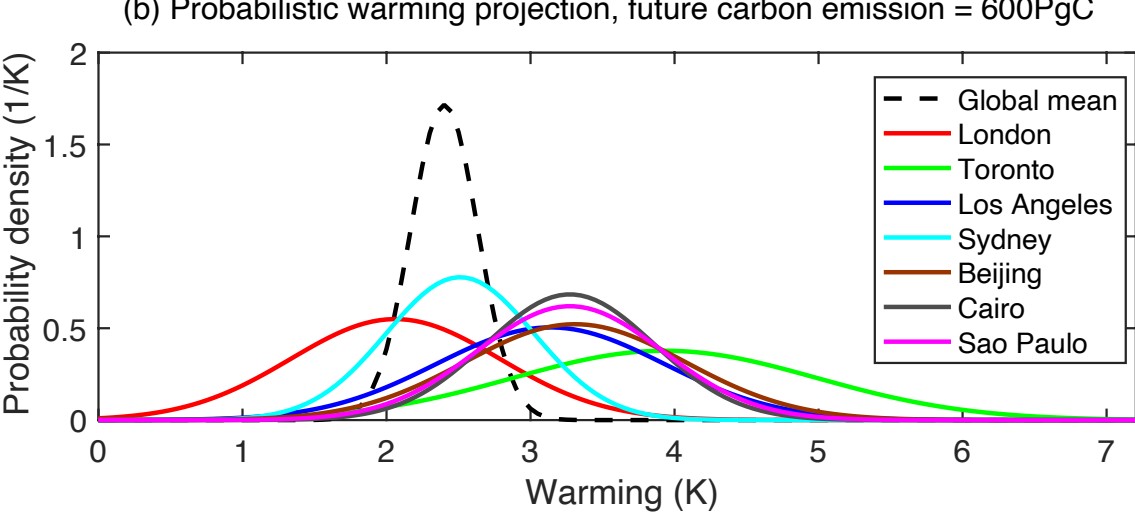

**Figure 5: Warming projections when future emissions reach 500 PgC from the start of 2018. (a) The spatial distribution of the central warming projection. (b) The probability distributions of local warming for 7 locations (solid colour lines) and the global surface average (black dashed line). All warming projections given relative to the average temperature from 1850 to 1900. Global mean warming projected from the quadratic approximation to the history matched WASP ensemble (eqns. 3 to 6) using the generic ≥2°C spatial pattern.**