# Peer review of "A computationally efficient method for probabilistic local warming projections constrained by history matching and pattern scaling, demonstrated by WASP-LGRTC-1.0"

_Geoscientific Model Development, 2019_

## Short Comment (SC1) · 15 Nov 2019

Dear authors,

in my role as Executive editor of GMD, I would like to bring to your attention our Editorial version 1.2:

https://www.geosci-model-dev.net/12/2215/2019/

This highlights some requirements of papers published in GMD, which is also available on the GMD website in the 'Manuscript Types' section:

http://www.geoscientific-model-development.net/submission/manuscript_types.html

[Figure]

In particular, please note that for your paper, the following requirement has not been met in the Discussions paper:

- "The main paper must give the model name and version number (or other unique identifier) in the title."

Please add the names/acronyms (WASP/LGRTC) of the models used/developed and their version numbers to the title upon your revised submission to GMD. Yours,

Astrid Kerkweg

---

## Referee Comment (RC1) · Christopher Smith (Referee) · 13 Dec 2019

General comments

This paper describes a simple methodology for translating global mean surface temperature diagnostic output from a simple climate model (WASP, but in theory any model like MAGICC, FAIR, Hector could theoretically be used) into regional surface temperature changes using a pattern scaling approach. While this is not a necessarily new concept (see fldgen: https://www.geosci-model-dev.net/12/1477/2019/), it is appreciated that a quick and simple tool would be greatly useful for translating the output of simple climate models (e.g. from IAMs) to regional impacts. Additionally, there is a

nice link from carbon emissions/carbon budgets to future carbon emissions. With this knowledge it could be possible to assess regional impacts as a function of the remaining carbon budget (e.g. to 1.5C).

Specific comments

∼Major

p4 l4-10: I am not sure if three scenarios that all show various rates of continually increasing warming are sufficient to make this conclusion. I would suspect that this does not hold for RCP2.6 where most models stabilise in temperature but regional patterns may continue to evolve. It would be good to show this. It would be helpful to see the 1pctCO2 scenarios for comparison in figure 1, also. (also relevant to p6 l6-10)

p4l22: a point on different non-CO2 forcers in the three scenarios - the RCPs are quite heterogeneous in their aerosol forcing in future scenarios, and 1pctCO2 does not include them. I'm not sure this gives us much information for pattern scaling for custom emissions scenarios. See figure 3 in Liu et al. for temperature responses to - admittedly somewhat extreme - cases of aerosol forcing in Europe and Asia. https://doi.org/10.1175/JCLI-D-17-0439.s1 . Some more discussion about how this model could handle widely varying timeseries of global and regional aerosol forcing would really help strengthen the model (and paper).

∼Minor

p1l36: a couple of years: I'd say it's more like 7 or 8, approximately in line with the corresponding IPCC Assessments.

p2l25: I think it's worth explaining which observational temperature datasets were used because the future projections you obtain from history matching will depend on whether they are observational blended global near-surface air temperautre and sea surface temperature and whether they infill for missing data.

Typographic/stylistic

various places: CO2: the 2 should be a subscript unless talking about named CMIP experiments where this is not the case (e.g. 1pctCO2).

p2l12: IMAGE and MESSAGE

p2l19: do you need to reiterate "efficient"?

p3l11: retracethe space missing

p4l9: not much relevant > not very relevant?

p4l39: observationally-constrained

---

## Referee Comment (RC2) · Anonymous Referee #2 · 10 Jan 2020

**1   General comments**

Goodwin et al. present a tool for projecting local warming with uncertainty from multiple anthropogenic emissions scenarios. The major advance of the paper is the combination of output from a probabilistic climate model and warming ratios from AOGCM/ESMs (I note that the MAGICC/SCENGEN, http://www.cgd.ucar.edu/cas/wigley/magicc/, tool does a similar thing but given that this paper is not tightly coupled to MAGICC or any other probabilistic climate model and its code is open sourced I consider this paper to be a significant advance on the MAG-ICC/SCENGEN tool). I feel that this advance could be a very useful addition to the

literature if a few concerns were addressed to provide more confidence in the paper's conclusions.

My major concerns focus on: whether the tool is actually scenario specific or not, how uncertainties from the climate model and LGRTC are combined and and whether WASP is actually a key part of the tool or whether any probabilistic climate model could be used.

One other key comment, given the availability of CMIP6 model output, I feel this paper could be significantly improved if it were to use CMIP6 output rather than focussing on CMIP5.

**2   Major concerns**

**2.1   Scenario specificity of pattern scaling**

It is not clear to me that the pattern scaling technique here is actually scenario agnostic. All the presented results are scenario specific (the RCP45 projections use RCP45 LGRTC and the RCP85 projections use RCP85 LGRTC) and there is no analysis of whether a 'general LGRTC' can be used nor whether such a 'general LGRTC' would have small enough uncertainties as to be useful.

I feel the comment (page 6, line 10), 'This allows future users to choose the spatial pattern scaling that is most suitable for their scenario.' is misleading. Only 3 patterns are available and none of them have been shown to be applicable for an emissions scenario different to the one from which they were derived (see comment above). Such cross-validation would be a vital step to providing confidence that the spatial pattern derived from one scenario can then be applied to any arbitrary scenario.

I am not convinced by the comment (page 4, line 8), 'The absolute value of differences

in LGRTC between the three scenarios was below 0.72C per°C in all grid-cells and mostly below 0.2C perC over the continents. Therefore, the choice of the emission scenario to define spatial pattern of warming in this study is not much relevant when only inhabited regions are considered.' Relative to strong mitigation targets (e.g. the 1.5°C target), I am not convinced these are trivial variations. In addition, in this context 'mostly' is meaningless and provides no quantification of how wide the disagreement is nor of the regions in which this generalisation doesn't hold (and how wrong it is).

I am also not convinced by the comment (page 4, line 19), 'This might have led to the large differences in the Arctic region, but detailed analysis and explanation is outside the scope of this study.' If the pattern scaling approach is to be used for arbitrary scenarios, there needs to be evidence that one pattern, with sufficiently large uncertainties, can be applied to multiple scenarios and give results that are in line with known results from CMIP models. Any differences need to be explained as they are of key interest when applying this tool (or the tools' domain of applicability should only be limited to those regions where the differences are small/well understood).

I think the data is there to address this concern. One suggestion (which would satisfy me) would be to derive some 'general LGRTC' (including uncertainty) which could be used with any emissions scenario. The 'general LGRTC' could then be applied to the RCPs (here meaning all RCPs, including RCP26 and RCP60, not just RCP45 and RCP85) and the differences quantified. This would provide a meaningful quantification of how big the uncertainties need to be on a 'general LGRTC' for it to sufficiently capture the variation across CMIP models and scenarios in the cases where we have data. I would be even more convinced if a 'general LGRTC' derived from CMIP5 RCPs was shown to hold for CMIP6 SSP scenarios.

**2.2 Scenario specificity of WASP**

WASP currently requires exogenous estimates of non-CO2 radiative forcing (see manuscript paragraph starting page 7, line 33). As far as I can tell, this means that this tool is not applicable to arbitray emissions scenarios but rather only ones for which there is an available non-CO2 radiative forcing quantification. I feel this is a rather fatal flaw of a tool which is meant to be applicable to arbitrary emissions scenarios.

An easy remedy would be to alter the tool from being 'WASP/LGRTC' to 'a general framework for coupling probabilistic climate model output and LGRTC' (insert acronym here) i.e. remove the explicit dependence on WASP. I can't see any reason why WASP is the only model with which this tool would work. This paper could still illustrate the use of the framework with WASP output, but such a reframing would make clear that the coupling could be done with any probabilistic climate model so a model which can run fully GHG-emissions driven could be used instead and would immediately fix the issue of WASP's limited available scenario set.

**2.3 Combination of uncertainties**

I am not convinced that the combination of uncertainties in equation 2 is correct. In equation 2, shouldn't the resulting distribution be the product/convolution of the two distributions rather than the output of random sampling from the two distributions? Given LGRTC is assumed to be gaussian, and that the WASP output is approximately gaussian, wouldn't it be better to derive the distribution of Delta T_i (x, y, t) by taking the product of two gaussians (see e.g. https://ccrma.stanford.edu/~jos/sasp/Product_Two_Gaussian_PDFs.html) which isn't the same as the product of two gaussian variables (see e.g. https://math.stackexchange.com/questions/101062/is-the-product-of-two-gaussian-random-variables-also-a-gaussian).

[Figure]

I'm happy to be corrected on this as I am not a statistical expert. However, regardless of whether I am correct or not I think some explanation must be added to the manuscript or the supplementary to explain this uncertainty propogation.

**2.4   Reliance on WASP**

It is not clear if this paper is using an existing WASP probabilistic distribution or presenting a new one (e.g. contradiction between page 5, line 9: '$3x10^6$ members' and page 2, line 23: '$10^8$ simulations'). If the reframing suggested earlier were to take place then this is no longer an issue (as the choice of particular probabilistic climate model is just for illustration and isn't a key feature of the tool). However, if this particular WASP probabilistic distribution is key then I would have to consider that component more closely.

(If the WASP probabilistic distribution is not key this entire paragraph can be ignored but for completeness) At the moment my only question is about the Monte Carlo sampling. Supplementary Table 2 of Goodwin et al. 2018b shows 18 parameters. With $3x10^6$ members you're effectively taking a bit over 5 steps in each parameter axis ($18^5$ ~ $2x10^6$). This appears to be a fairly sparse sampling, which could be a problem no? I wasn't convinced by Goodwin et al. 2018b, 'This observation-consistent ensemble displays good agreement with the full ranges for all the observational quantities (Supplementary Table 4), which demonstrates that the $3x10^4$ simulations have a good coverage of observational parameter space.' It seems perfectly plausible to me that the 95% ranges could agree but the distributions themselves are otherwise very different. If you've considered this before and can include the answers in the paper or point to them in the paper that would be great, if not then a sentence highlighting this and saying that they're areas for future research would suffice.

**3  Specific comments**

1. 'Thus a TCRE framework is applicable for certain situations, including idealised scenarios where the TRCE has already been established, but in the general case a time-dependent Earth system model is required.' (page 2, line 13) Can you make some comment about what this means for the TCRE framework that was heavily used in SR1.5 (see Rogelj et al. Nature 2019, 'Estimating and tracking the remaining carbon budget for stringent climate targets')? For example, does it mean that the framework can only be applied if its components were derived with a suitable scenario set?

2. Opening paragraph, the commas mean that the sentence says 'The dominant climate projections are made using the Climate Model Inter-comparison Project phase 5 ensemble'. Given CMIP6 is well and truly underway, can you re-write the entire paragraph to make CMIP references more general e.g. 'The dominant climate projections are made using results from multiple phases of the Coupled Model Inter-comparison Project' (the second sentence of the paragraph also needs a similar adjustment)? (See also comment about using CMIP6 data throughout the manuscript)

3. 'ideal tool for future incorporation into an Integrated Assessment Model framework' (page 1, line 25). Given WASP's requirement of exogenous non-CO2 forcing I don't think this is true so would remove this phrase. It could, of course, remain if the reframing towards a more general probabilistic climate model-LGRTC framework suggested above was made.

4. The paragraph beginning page 4, line 11. I was very confused by the entire discussion of reference periods and comparison periods throughout this section. Specifically, 'we have chosen the preindustrial climate as the reference period in 1pctCO2'. What does this mean? Is the reference period in 1pctCO2 1850-1900

or something else? 'in the RCP scenarios we used beginning of the 21st century'. Is my understanding right that you are saying you used the beginning of the 21st century as the pre-industrial climate for the RCPs? If yes, this seems a very odd choice. 'end period... years 2079-2098 in the RCPs', this seems a very odd choice of end period, why not 2081-2100 as is used in the IPCC reports?

5. why are the TCRE fits a) quadratic and b) only done to one scenario? Given the use of the TCRE concept throughout the literature, either a linear, scenario-independent fit between warming and cumulative carbon emissions should be done or a much more thorough discussion of why a scenario-dependent, quadratic fit is appropriate should be added.

**4  Figures**

Figure 1 caption: is this the multi-model mean?

Figure 2: a different colour palette and increments would be helpful so you can see whether the standard deviation is in 0-0.3 or 0.3-0.6, very hard to tell at the moment and such a difference are of interest with respect to the renewed focus on 0.5C temperature increments following SR1.5

Figure 4: add standard deviation panel too please so the size of the uncertainties is immediately obvious (doing the differences by eye is basically impossible given how wide the colour bar scale is)

**5  Technical corrections**

page 1, line 24: delete 'arbitrary'

page 1, line 27: 'tool making' → 'tool for making'

page 1, line 31: 'CMIP' stands for 'Coupled Model Intercomparison Project', not 'Climate Model Intercomparison Project'

page 1, line 34: 'carbon-emissions' → 'anthropogenic-emission'

page 2, line 2: 'resolutionand/ora' → 'resolution and/or a'

page 2, line 6: 'and an' → 'and a'

page 2, line 6: 'hemispherical averaged' → 'hemispherical land-ocean averaged'

page 2, line 9: 'is applied' → 'can be applied'

page 2, line 10: do you have a reference for SCENGEN? Perhaps http://www.cgd.ucar.edu/cas/wigley/magicc/?

page 2, line 12: 'IMAGE, MESSAGEframeworks' → 'IMAGE and MESSAGE frameworks'

page 3, line 11-12: 'for example where cooling following negative emissions may not re-tracethe previous warming pathway' needs to be re-written, I can guess what you're saying but the sentence doesn't actually make sense

page 4, line 5: delete 'such'

page 4, line 11: 'Despite that' → 'Despite the fact that' ? I'm not actually sure what is intended here.

page 4, line 22: 'RCP scenarios more usable than those from the 1pctCO2 scenario to be used to predict warming patterns in the 21st century' → 'RCP scenarios more appropriate those from the 1pctCO2 scenario to predict warming patterns in the 21st century'

page 4, line 27: Missing units on the numbers

page 4, line 30: This paragraph is a very roundabout way of saying that the projections from each CMIP model are internally physically consistent, but as the WASP/LGRTC uses averages of each CMIP model, its results are unlikely to be internally physically consistent. Can you cut the paragraph to one or two sentences?

page 4, line 39: Delete this paragraph

page 5, line 17: ', (Figure 3)' → ' (Figure 3)'

page 6, line 15: 'function' → 'a function'

page 7, line 17: Calling WASP an 'efficient ESM' is a stretch, use 'efficient climate model' or 'simple ESM' or 'ESM emulator' instead

page 7, line 23: Straw-manning paragraph, the WASP part is not new (lots of other emulators with probabilistic distributions constrained to history), only the LGRTC scaling is

page 7, line 38: Given lack of any tests for other platforms or docs for how to set things up, I am dubious of the phrase, 'Both the WASP/LGRTC model and the quadratic approximation to WASP/LGRTC model are easy to use', particularly in relation to WASP's useability. Given all the plotting is all MATLAB based you have a significant barrier to entry.

**5.1 Supplementary information**

page 3, Table caption: Am I correct in guessing that the '90 to 95%' confidence interval means that 95% confidence interval was used where data was available otherwise 90% confidence interval was used? If not, can you please clarify what a range of confidence interval means as I haven't seen such a usage before (I would have thought it would just be '95% confidence interval')?

---

## Editor Comment (EC1) · James Annan (Editor) · 16 Jan 2020

The reviewer contacted me with this lightly edited version of his original review, which should be taken as replacing the original. Reviewer comments start immediately:

**1   General comments**

Goodwin et al. present a tool for projecting local warming with uncertainty from multiple anthropogenic emissions scenarios.    The major advance of

the paper is the combination of output from a probabilistic climate model and warming ratios from AOGCM/ESMs (I note that the MAGICC/SCENGEN, http://www.cgd.ucar.edu/cas/wigley/magicc/, tool does a similar thing but given that this paper is not tightly coupled to MAGICC or any other probabilistic climate model and its code is open sourced I consider this paper to be a significant advance on the MAG-ICC/SCENGEN tool). I feel that this advance could be a very useful addition to the literature if a few concerns were addressed to provide more confidence in the paper's conclusions.

My major concerns focus on: whether the tool is actually scenario specific or not, how uncertainties from the climate model and LGRTC are combined and and whether WASP is actually a key part of the tool or whether any probabilistic climate model could be used.

One other key comment, given the availability of CMIP6 model output, I feel this paper could be significantly improved if it were to use CMIP6 output rather than focussing on CMIP5.

**2  Major concerns**

**2.1  Scenario specificity of pattern scaling**

It is not clear to me that the pattern scaling technique here is actually scenario agnostic. All the presented results are scenario specific (the RCP45 projections use RCP45 LGRTC and the RCP85 projections use RCP85 LGRTC) and there is no analysis of whether a 'general LGRTC' can be used nor whether such a 'general LGRTC' would have small enough uncertainties as to be useful.

I feel the comment (page 6, line 10), 'This allows future users to choose the spatial

pattern scaling that is most suitable for their scenario.' is misleading. Only 3 patterns are available and none of them have been shown to be applicable for an emissions scenario different to the one from which they were derived (see comment above). Such cross-validation would be a vital step to providing confidence that the spatial pattern derived from one scenario can then be applied to any arbitrary scenario.

I am not convinced by the comment (page 4, line 8), 'The absolute value of differences in LGRTC between the three scenarios was below 0.72C per°C in all grid-cells and mostly below 0.2C perC over the continents. Therefore, the choice of the emission scenario to define spatial pattern of warming in this study is not much relevant when only inhabited regions are considered.' Relative to strong mitigation targets (e.g. the 1.5°C target), I am not convinced these are trivial variations. In addition, in this context 'mostly' is meaningless and provides no quantification of how wide the disagreement is nor of the regions in which this generalisation doesn't hold (and how wrong it is).

I am also not convinced by the comment (page 4, line 19), 'This might have led to the large differences in the Arctic region, but detailed analysis and explanation is outside the scope of this study.' If the pattern scaling approach is to be used for arbitrary scenarios, there needs to be evidence that one pattern, with sufficiently large uncertainties, can be applied to multiple scenarios and give results that are in line with known results from CMIP models. Any differences need to be explained as they are of key interest when applying this tool (or the tools' domain of applicability should only be limited to those regions where the differences are small/well understood).

I think the data is there to address this concern. One suggestion (which would satisfy me) would be to derive some 'general LGRTC' (including uncertainty) which could be used with any emissions scenario. The 'general LGRTC' could then be applied to the RCPs (here meaning all RCPs, including RCP26 and RCP60, not just RCP45 and RCP85) and the differences quantified. This would provide a meaningful quantification of how big the uncertainties need to be on a 'general LGRTC' for it to sufficiently capture the variation across CMIP models and scenarios in the cases where we have data. I

would be even more convinced if a 'general LGRTC' derived from CMIP5 RCPs was shown to hold for CMIP6 SSP scenarios.

**2.2   Scenario specificity of WASP**

WASP currently requires exogenous estimates of non-CO2 radiative forcing (see manuscript paragraph starting page 7, line 33). As far as I can tell, this means that this tool is not applicable to arbitray emissions scenarios but rather only ones for which there is an available non-CO2 radiative forcing quantification. I feel this is a rather fatal flaw of a tool which is meant to be applicable to arbitrary emissions scenarios.

An easy remedy would be to alter the tool from being 'WASP/LGRTC' to 'a general framework for coupling probabilistic climate model output and LGRTC' (insert acronym here) i.e. remove the explicit dependence on WASP. I can't see any reason why WASP is the only model with which this tool would work. This paper could still illustrate the use of the framework with WASP output, but such a reframing would make clear that the coupling could be done with any probabilistic climate model so a model which can run fully GHG-emissions driven could be used instead and would immediately fix the issue of WASP's limited available scenario set.

**2.3   Combination of uncertainties**

I am not convinced that the combination of uncertainties in equation 2 is correct. In equation 2, shouldn't the resulting distribution be the product/convolution of the two distributions rather than the output of random sampling from the two distributions?   Given LGRTC is assumed to be gaussian, and that the WASP output is approximately gaussian, wouldn't it be better to derive the distribution of Delta T_i (x, y, t) by taking the product of two gaussians (see e.g.   https://ccrma.stanford.edu/~jos/sasp/Product_Two_Gaussian_PDFs.html)

which isn't the same as the product of two gaussian variables (see e.g. https://math.stackexchange.com/questions/101062/is-the-product-of-two-gaussian-random-variables-also-a-gaussian).

I'm happy to be corrected on this as I am not a statistical expert. However, regardless of whether I am correct or not I think some explanation must be added to the manuscript or the supplementary to explain this uncertainty propogation.

**2.4 Reliance on WASP**

It is not clear if this paper is using an existing WASP probabilistic distribution or presenting a new one (e.g. contradiction between page 5, line 9: '$3\times10^6$ members' and page 2, line 23: '$10^8$ simulations'). If the reframing suggested earlier were to take place then this is no longer an issue (as the choice of particular probabilistic climate model is just for illustration and isn't a key feature of the tool). However, if this particular WASP probabilistic distribution is key then I would have to consider that component more closely.

(If the WASP probabilistic distribution is not key this entire paragraph can be ignored but for completeness) At the moment my only question is about the Monte Carlo sampling. Supplementary Table 2 of Goodwin et al. 2018b shows 18 parameters. With $3\times10^6$ members you're effectively taking a bit over 5 steps in each parameter axis ($18^5 \sim 2\times10^6$). This appears to be a fairly sparse sampling, which could be a problem no? I wasn't convinced by Goodwin et al. 2018b, 'This observation-consistent ensemble displays good agreement with the full ranges for all the observational quantities (Supplementary Table 4), which demonstrates that the $3\times10^4$ simulations have a good coverage of observational parameter space.' It seems perfectly plausible to me that the 95% ranges could agree but the distributions themselves are otherwise very different. If you've considered this before and can include the answers in the paper or point to them in the paper that would be great, if not then a sentence highlighting this and

saying that they're areas for future research would suffice.

**3 Specific comments**

1. 'Thus a TCRE framework is applicable for certain situations, including idealised scenarios where the TRCE has already been established, but in the general case a time-dependent Earth system model is required.' (page 2, line 13) Can you make some comment about what this means for the TCRE framework that was heavily used in SR1.5 (see Rogelj et al. Nature 2019, 'Estimating and tracking the remaining carbon budget for stringent climate targets')? For example, does it mean that the framework can only be applied if its components were derived with a suitable scenario set?

2. Opening paragraph, the commas mean that the sentence says 'The dominant climate projections are made using the Climate Model Inter-comparison Project phase 5 ensemble'. Given CMIP6 is well and truly underway, can you re-write the entire paragraph to make CMIP references more general e.g. 'The dominant climate projections are made using results from multiple phases of the Coupled Model Inter-comparison Project' (the second sentence of the paragraph also needs a similar adjustment)? (See also comment about using CMIP6 data throughout the manuscript)

3. 'ideal tool for future incorporation into an Integrated Assessment Model framework' (page 1, line 25). Given WASP's requirement of exogenous non-CO2 forcing I don't think this is true so would remove this phrase. It could, of course, remain if the reframing towards a more general probabilistic climate model-LGRTC framework suggested above was made.

4. The paragraph beginning page 4, line 11. I was very confused by the entire

discussion of reference periods and comparison periods throughout this section. Specifically, 'we have chosen the preindustrial climate as the reference period in 1pctCO2'. What does this mean? Is the reference period in 1pctCO2 1850-1900 or something else? 'in the RCP scenarios we used beginning of the 21st century'. Is my understanding right that you are saying you used the beginning of the 21st century as the pre-industrial climate for the RCPs? If yes, this seems a very odd choice. 'end period... years 2079-2098 in the RCPs', this seems a very odd choice of end period, why not 2081-2100 as is used in the IPCC reports?

5. why are the TCRE fits a) quadratic and b) only done to one scenario? Given the use of the TCRE concept throughout the literature, either a linear, scenario-independent fit between warming and cumulative carbon emissions should be done or a much more thorough discussion of why a scenario-dependent, quadratic fit is appropriate should be added.

6. In my opinion, the paragraph beginning page 7, line 21 is wrong and it should be removed in its entirety. The tool is not an alternative to existing simple climate models (as far as I'm aware no other tool with this downscaling coupling exists except MAGICC/SCENGEN, and that has already been covered in this review) but rather presents a new way of coupling simple model output and LGRTC. There are already lots of simple models with probabilistic distributions (MAGICC see Meinshausen et al Nature 2009, FaIR, CICERO-SCM, OSCAR, Hector...) so chosing to only compare WASP and the emulations from Meinshausen et al 2011, rather than the existing probabilistic sets, is completely unjustified.

**4  Figures**

Figure 1 caption: is this the multi-model mean?

**GMDD**

Figure 2: a different colour palette and increments would be helpful so you can see whether the standard deviation is in 0-0.3 or 0.3-0.6, very hard to tell at the moment and such a difference are of interest with respect to the renewed focus on 0.5C temperature increments following SR1.5

Figure 4: add standard deviation panel too please so the size of the uncertainties is immediately obvious (doing the differences by eye is basically impossible given how wide the colour bar scale is)

**5  Technical corrections**

page 1, line 24: delete 'arbitrary'

page 1, line 27: 'tool making' → 'tool for making'

page 1, line 31: 'CMIP' stands for 'Coupled Model Intercomparison Project', not 'Climate Model Intercomparison Project'

page 1, line 34: 'carbon-emissions' → 'anthropogenic-emission'

page 2, line 2: 'resolutionand/ora' → 'resolution and/or a'

page 2, line 6: 'and an' → 'and a'

page 2, line 6: 'hemispherical averaged' → 'hemispherical land-ocean averaged'

page 2, line 9: 'is applied' → 'can be applied'

page 2, line 10: do you have a reference for SCENGEN? Perhaps http://www.cgd.ucar.edu/cas/wigley/magicc/?

page 2, line 12: 'IMAGE, MESSAGEframeworks' → 'IMAGE and MESSAGE frameworks'

page 3, line 11-12: 'for example where cooling following negative emissions may not re-tracethe previous warming pathway' needs to be re-written, I can guess what you're saying but the sentence doesn't actually make sense

page 4, line 5: delete 'such'

page 4, line 11: 'Despite that' → 'Despite the fact that' ? I'm not actually sure what is intended here.

page 4, line 22: 'RCP scenarios more usable than those from the 1pctCO2 scenario to be used to predict warming patterns in the 21st century' → 'RCP scenarios more appropriate those from the 1pctCO2 scenario to predict warming patterns in the 21st century'

page 4, line 27: Missing units on the numbers

page 4, line 30: This paragraph is a very roundabout way of saying that the projections from each CMIP model are internally physically consistent, but as the WASP/LGRTC uses averages of each CMIP model, its results are unlikely to be internally physically consistent. Can you cut the paragraph to one or two sentences?

page 4, line 39: Delete this paragraph

page 5, line 17: ', (Figure 3)' → ' (Figure 3)'

page 6, line 15: 'function' → 'a function'

page 7, line 17: Calling WASP an 'efficient ESM' is a stretch, use 'efficient climate model' or 'simple ESM' or 'ESM emulator' instead

page 7, line 38: Given that the code isn't subject to any testing except for manual validation on the authors' machines (at least that is how it appears from looking at the GitHub repository), I am dubious of the phrase, 'Both the WASP/LGRTC model and the quadratic approximation to WASP/LGRTC model are easy to use'. Particularly in relation to WASP's useability, all the plotting is MATLAB

based which is a significant barrier to entry and the setup has to re-run the entire $3x10^{6} parameter sets every time which doesn't give the impression of being easy. I would delete this sentence.$

**5.1 Supplementary information**

page 3, Table caption: Am I correct in guessing that the '90 to 95%' confidence interval means that 95% confidence interval was used where data was available otherwise 90% confidence interval was used? If not, can you please clarify what a range of confidence interval means as I haven't seen such a usage before (I would have thought it would just be '95% confidence interval')?

---

## Author Comment (AC1) · 12 Feb 2020

**Reply to SC1: Authors' response to the editorial comment:**

The comment by Astrid Kerkweg drew attention to an editorial requirement:

**Reviewer's comment:** *"In particular, please note that for your paper, the following requirement has not been met in the Discussions paper:*

*"The main paper must give the model name and version number (or other unique identifier) in the title."*

*Please add the names/acronyms (WASP/LGRTC) of the models used/developed and*

[Figure]

*their version numbers to the title upon your revised submission to GMD. Yours,*

*Astrid Kerkweg"*

**Authors' response:** Thank you for drawing this editorial requirement to our attention. When revising the manuscript, we will include in the manuscript title a unique model name and version number for the combined WASP/LGRTC model for spatial warming patterns.

**Reply to RC1: Authors' responses to reviewer 1's comments:**

We thank reviewer 1, Dr. Christopher Smith, for important and insightful comments about our manuscript. Below we explain how we will use these comments to further improve our manuscript.

**Reviewer's comment:** *"General comments This paper describes a simple methodology for translating global mean surface temperature diagnostic output from a simple climate model (WASP, but in theory any model like MAGICC, FAIR, Hector could theoretically be used) into regional surface temperature changes using a pattern scaling approach. While this is not a necessarily new concept (see fldgen: https://www.geosci-model-dev.net/12/1477/2019/), it is appreciated that a quick and simple tool would be greatly useful for translating the output of simple climate models (e.g. from IAMs) to regional impacts. Additionally, there is a nice link from carbon emissions/carbon budgets to future carbon emissions. With this knowledge it could be possible to assess regional impacts as a function of the remaining carbon budget (e.g. to 1.5C)."*

**Authors' response: Point 1: Applicability of the spatial tool to any model capable of projecting global mean warming.** We agree that any model capable of generating probabilistic projections for global mean surface temperature could be combined with the spatial tools presented in this manuscript. This would then generate projections for the mean and standard deviation for future local warming. In a revised manuscript we will highlight how the spatial tool can be coupled to any model projecting global mean

warming, and not just the WASP model. This change will make the manuscript more general in nature and of interest to a wider range of readers.

**Point 2: Link to emissions budgets and regional impacts.** We agree that the approximation tool presented provides a useful link to assess regional impacts from carbon emissions/carbon budgets. In a revised manuscript we will improve this link by exploring the spatial warming pattern (the LGRTC) for the RCP2.6 scenario – a scenario with strong mitigation and a high likelihood of meeting the Paris Climate Agreement goals.

**Reviewer's comment:** *"p4 l4-10: I am not sure if three scenarios that all show various rates of continually increasing warming are sufficient to make this conclusion. I would suspect that this does not hold for RCP2.6 where most models stabilise in temperature but regional patterns may continue to evolve. It would be good to show this. It would be helpful to see the 1pctCO2 scenarios for comparison in figure 1, also. (also relevant to p6 l6-10)"*

**Authors' response: Scenarios with increasing warming and scenarios with stabilised warming.** We agree that the scenarios considered have increasing warming although we note that in RCP4.5 the there is little additional warming after 2080 across the 13 CMIP5 models considered by Goodwin et al., (2018b – see figure 2c therein, grey shaded area), we agree that there is little time for the warming pattern to continue to evolve.

We also agreed that the RCP2.6 scenario offers a chance to explore our LGRTC tool for generating local warming projections in a scenario with stabilised climate. In a revised version we will produce a LGRTC for RCP2.6 and compare this to the existing scenarios. This will allow us to test our methodology for a climate with warming stabilised close to the Paris Climate Agreement warming targets of 1.5 and 2.0 °C.

**Reviewer's comment:** *"p4l22: a point on different non-CO2 forcers in the three scenarios - the RCPs are quite heterogeneous in their aerosol forcing in future scenarios,*

*and 1pctCO2 does not include them. I'm not sure this gives us much information for pattern scaling for custom emissions scenarios. See figure 3 in Liu et al. for temperature responses to - admittedly somewhat extreme - cases of aerosol forcing in Europe and Asia. https://doi.org/10.1175/JCLI-D-17-0439.s1 . Some more discussion about how this model could handle widely varying timeseries of global and regional aerosol forcing would really help strengthen the model (and paper)."*

**Authors' response: Pattern scaling for custom emissions scenarios with extreme aerosol emissions.** In the production of the RCP scenarios, assumptions have been made as to the relative amounts of different forcing agents emitted (e.g. $CO_2$, other well mixed greenhouse gasses, aerosols etc) (e.g. see IPCC, 2013 or Meinshausen et al., 2011). Some forcing agents, specifically aerosols, are not well mixed in the atmosphere but exert a significant local influence. We agree that for extreme cases of localised aerosol radiative forcing the pattern of warming would be different to the assumptions used in the generation of the different RCP scenarios.

In a revised manuscript, we will discuss how this approach could be extended in future study to account for custom scenarios containing extreme aerosols forcing. The work itself is reserved for future study because, fundamentally, one needs additional information from complex climate models involving separate model integrations with non RCP (or SSP) style scenarios. Briefly, a revised manuscript will discuss how the method presented could be extended in future work by: (1) Calculating the LGRTC for a scenario with $CO_2$ only (or well mixed greenhouse gas only) forcing, (2) Exploring the sensitivity of LGRTC patterns to regional aerosol emission in complex climate models, for cases with idealised aerosol emissions for each region (e.g. Europe, Asia etc). (3) Combining the LGRTC from well mixed greenhouse forcing with the LGRTC from idealised aerosol forcing in each region, using knowledge of the relative emissions from greenhouse gasses and aerosols by region in the custom scenario, to generate a custom warming pattern for a scenario with extreme aerosol emission patterns. The key to such a method working is that the uncertainties introduced by the assumptions involved

in combining different spatial patterns are smaller than the uncertainties introduced by the range of CMIP class model responses for a given scenario.

Since this method relies on new targeted experiments with complex CMIP models, this is not feasible to conduct for this paper. However, we will spell out the possible methods by which this could be achieved in future work in the revised version.

Note that the calculations for the LGRTC presented here will apply for custom scenarios that do utilise similar assumptions to the RCP scenarios in terms of the relative amounts of well mixed greenhouse emissions and aerosols.

Minor comments:

**Reviewer comment:** *"Minor p1l36: a couple of years: I'd say it's more like 7 or 8, approximately in line with the corresponding IPCC Assessments."*

**Authors' response:** Agreed, we a revised manuscript will be amended accordingly.

**Reviewer's comment:** *"p2l25: I think it's worth explaining which observational temperature datasets were used because the future projections you obtain from history matching will depend on whether they are observational blended global near-surface air temperautre and sea surface temperature and whether they infill for missing data."*

**Authors' response:** Agreed – the global mean temperature datasets used are blended land air temperature and sea surface temperature records (e.g. HadCRUT4 and GISTEMP). We will spell this out in a revised manuscript so that readers understand precisely what our global mean temperature anomaly projections should be compared to.

**Typographic/stylistic points.** We also thank Dr. Smith for also raising minor/stylistic points. We confirm that we agree with all minor/stylistic comments raised by reviewer 1 and can confirm that these will all be addressed in a revised version.

**Reply to RC2 (as updated in EC1): Reply to reviewer 2's comments**

**Reviewer comment:** *"1 General comments Goodwin et al. present a tool for projecting*

[Figure]

*local warming with uncertainty from multiple anthropogenic emissions scenarios. The major advance of the paper is the combination of output from a probabilistic climate model and warming ratios from AOGCM/ESMs (I note that the MAGICC/SCENGEN, http://www.cgd.ucar.edu/cas/wigley/magicc/, tool does a similar thing but given that this paper is not tightly coupled to MAGICC or any other probabilistic climate model and its code is open sourced I consider this paper to be a significant advance on the MAG-ICC/SCENGEN tool). I feel that this advance could be a very useful addition to the literature if a few concerns were addressed to provide more confidence in the paper's conclusions."*

**Authors' response: General comments.** We are pleased the reviewer sees the advance offered, and in a revision we will amend the manuscript to address the specific concerns – please see details below.

**Reviewer comment:** *"My major concerns focus on: whether the tool is actually scenario specific or not, how uncertainties from the climate model and LGRTC are combined and and whether WASP is actually a key part of the tool or whether any probabilistic climate model could be used.*

*"One other key comment, given the availability of CMIP6 model output, I feel this paper could be significantly improved if it were to use CMIP6 output rather than focussing on CMIP5."*

**Authors' response: Concerns.** We spell out in detail below how we will update a revised manuscript to address the concerns raised. In brief, in a revised manuscript we will: (1) Analyse the LGRTC for an additional scenario, RCP2.6, and provide more robust statistical comparisons of the differences in LGRTC for the different scenarios, including identifying a spatial domain over which a single LGRTC can be applied; (2) Stress that the method is not specific to the WASP model, and make it clear that our methodology can be applied to any efficient model generating projections of global mean surface warming; and (3) We will reserve the analysis of CMIP6 model output for

future study.

**Reviewer comment:** *"2.1 Scenario specificity of pattern scaling*

*"It is not clear to me that the pattern scaling technique here is actually scenario agnostic. All the presented results are scenario specific (the RCP45 projections use RCP45 LGRTC and the RCP85 projections use RCP85 LGRTC) and there is no analysis of whether a 'general LGRTC' can be used nor whether such a 'general LGRTC' would have small enough uncertainties as to be useful."*

**Authors' response: Scenario specificity of the LGRTC.** Agreed that an analysis of whether a 'general LGRTC' can be defined will improve the manuscript. We will include such an analysis in the revised version (see below for details)

**Reviewer comment:** *"I feel the comment (page 6, line 10), 'This allows future users to choose the spatial pattern scaling that is most suitable for their scenario.' is misleading. Only 3 patterns are available and none of them have been shown to be applicable for an emissions scenario different to the one from which they were derived (see comment above). Such cross-validation would be a vital step to providing confidence that the spatial pattern derived from one scenario can then be applied to any arbitrary scenario."*

**Authors' response:** Agreed that the sentence "This allows future users to choose the spatial pattern scaling that is most suitable for their scenario' is unclear in its present form. This statement will be removed in a revised version. See below for details on broader points.

**Reviewer comment:** *"I am not convinced by the comment (page 4, line 8), 'The absolute value of differences in LGRTC between the three scenarios was below 0.72C perC in all grid-cells and mostly below 0.2C perC over the continents. Therefore, the choice of the emission scenario to define spatial pattern of warming in this study is not much relevant when only inhabited regions are considered.' Relative to strong mitigation targets (e.g. the 1.5C target), I am not convinced these are trivial variations. In addition,*

[Figure]

*in this context 'mostly' is meaningless and provides no quantification of how wide the disagreement is nor of the regions in which this generalisation doesn't hold (and how wrong it is)."*

*"I am also not convinced by the comment (page 4, line 19), 'This might have led to the large differences in the Arctic region, but detailed analysis and explanation is outside the scope of this study.' If the pattern scaling approach is to be used for arbitrary scenarios, there needs to be evidence that one pattern, with sufficiently large uncertainties, can be applied to multiple scenarios and give results that are in line with known results from CMIP models. Any differences need to be explained as they are of key interest when applying this tool (or the tools' domain of applicability should only be limited to those regions where the differences are small/well understood)."*

**Authors' response: Imprecise wording of comparisons between scenario-LGRTC patters.** Agreed that the highlighted sentences do not provide robust statistical analysis of the differences and similarities between the LGRTC patterns for the scenarios. In a revised manuscript, we will provide a robust statistical comparison of the LGRTC for different scenarios over different areas of the domain. We will define the domain over which the tool is applicable. See answer to the next paragraph for more details.

**Reviewer comment:** *"I think the data is there to address this concern. One suggestion (which would satisfy me) would be to derive some 'general LGRTC' (including uncertainty) which could be used with any emissions scenario. The 'general LGRTC' could then be applied to the RCPs (here meaning all RCPs, including RCP26 and RCP60, not just RCP45 and RCP85) and the differences quantified. This would provide a meaningful quantification of how big the uncertainties need to be on a 'general LGRTC' for it to sufficiently capture the variation across CMIP models and scenarios in the cases where we have data. I would be even more convinced if a 'general LGRTC' derived from CMIP5 RCPs was shown to hold for CMIP6 SSP scenarios."*

**Authors response: Addressing concern over applicability of LGRTC for different scenarios/scenario dependence quantification.** Agreed that greater insight into the amount of scenario-dependence is required, and that this affects the validity of using the LGRTC for scenarios other than the scenarios from which they were derived. In a revised manuscript we will:

(1) Analyse the LGRTC for an additional scenario, RCP2.6 (a stabilisation scenario with strong mitigation in line with the Paris Climate Agreement's targets of keeping warming under 2.0 °C).

(2) Provide a more meaningful statistical comparison of the LGRTC for the different scenarios (including RCP8.5, RCP4.5, RCP2.6 and 1 per cent CO2), including comparing the magnitude of uncertainty due to differences 'within scenario but between CMIP5 models' to the differences 'between scenarios'. i.e. comparing sigma_LGRTC within a scenario to the differences between mu_LGRTC for different scenarios.

This comparison will avoid language such as 'mostly' and be quantitative as to the differences between scenarios over various spatial domains.

(3) Explore the feasibility of defining a domain over which a general LGRTC can be defined (with uncertainties large enough to capture variation across CMIP5 models and variation between scenarios).

Ultimately, a key property of a single LGRTC must be that the uncertainty introduced by the scenario choice is less than the uncertainty introduced by the range of CMIP-class model responses within each a given scenario. In a revised manuscript, we will identify domains over a single LGRTC can be identified for all scenarios.

We will reserve comparisons to CMIP6 for future study.

**Reviewer's comment:** *"2.2 Scenario specificity of WASP*

*WASP currently requires exogenous estimates of non-CO2 radiative forcing (see manuscript paragraph starting page 7, line 33). As far as I can tell, this means that*

[Figure]

*this tool is not applicable to arbitray emissions scenarios but rather only ones for which there is an available non-CO2 radiative forcing quantification. I feel this is a rather fatal flaw of a tool which is meant to be applicable to arbitrary emissions scenarios.*

*An easy remedy would be to alter the tool from being 'WASP/LGRTC' to 'a general framework for coupling probabilistic climate model output and LGRTC' (insert acronym here) i.e. remove the explicit dependence on WASP. I can't see any reason why WASP is the only model with which this tool would work. This paper could still illustrate the use of the framework with WASP output, but such a reframing would make clear that the coupling could be done with any probabilistic climate model so a model which can run fully GHG-emissions driven could be used instead and would immediately fix the issue of WASP's limited available scenario set."*

**Authors' response: The method's (non)reliance on WASP.** We agree that the LGRTC tool can be applied to any arbitrary probabilistic climate model ensemble, not just the WASP ensemble used in the study. We will reframe the manuscript in terms of offering a general framework, with WASP the efficient model used to illustrate the tool.

**Reviewer's comment:** *"2.3 Combination of uncertainties*

*I am not convinced that the combination of uncertainties in equation 2 is cor-rect. In equation 2, shouldn't the resulting distribution be the product/convolution of the two distributions rather than the output of random sampling from the two distributions? Given LGRTC is assumed to be gaussian, and that the WASP output is approximately gaussian, wouldn't it be better to derive the distribution of Delta T_i (x, y, t) by taking the product of two gaussians (see e.g. https://ccrma.stanford.edu/ jos/sasp/Product_Two_Gaussian_PDFs.html) which isn't the same as the product of two gaussian variables (see e.g. https://math.stackexchange.com/questions/101062/is-the-product-of-twogaussian-random-variables-also-a-gaussian). I'm happy to be corrected on this as I am not a statistical expert. However, regardless of whether I am correct or not I think some*

*explanation must be added to the manuscript or the supplementary to explain this uncertainty propogation."*

**Authors' response: Combination of uncertainties.** We agree with the statistical points made about the random sampling of two Gaussian distributions not in general giving the same answer as the convolution of two Gaussian distributions.

However, to make our LGRTC method applicable to any arbitrary probabilistic projection of global mean surface warming (not just from this WASP ensemble), we cannot assume that the projection of global mean surface warming is Gaussian. Therefore, to ensure that our approach is generally applicable to any efficient model's projection of global mean surface warming, we cannot take product of two Gaussian distributions as suggested by the reviewer.

Our method, of randomly sampling from both the distributions of global mean warming and LGRTC, is applicable to any arbitrary projection of global mean surface warming.

We also point out that in the MATLAB approximation tool, which does tie in to the WASP ensemble, we have used the product of two Gaussian distributions rather than random sampling.

In revision, we will clarify why we cannot assume a Gaussian distribution for future global mean surface warming if our approach local warming approach is to be applied to any arbitrary model for generating probabilistic projections of global mean warming. We will also state that it is an option to convolute the Gaussian distributions, in the special case where one already knows that the global mean warming distribution is Gaussian.

**Reviewer's comment:** *"2.4 Reliance on WASP It is not clear if this paper is using an existing WASP probabilistic distribution or presenting a new one (e.g. contradiction between page 5, line 9: '3x10$^6$ members' and page 2, line 23: '10$^8$ simulations'). If the reframing suggested earlier were to take place then this is no longer an issue (as*

[Figure]

*the choice of particular probabilistic climate model is just for illustration and isn't a key feature of the tool). However, if this particular WASP probabilistic distribution is key then I would have to consider that component more closely."*

**Authors' response: The (non)reliance on WASP.** The novel methodology (of combining the LGRTC with a probabilistic ensemble of global mean warming from an efficient numerical model) is not tied to WASP. Therefore, we will be making the reframing suggested by the reviewer earlier clear in a revised manuscript. We adopt the probabilistic ensemble generated in Goodwin et al (2018b). We will make the particular ensemble used clear in the revised manuscript.

We note that there is not a contradiction between the $3 \times 10^6$ members of the posterior ensemble and $10^8$ members of the prior ensemble in the WASP methodology (see below).

**Reviewer's comment:** *"(If the WASP probabilistic distribution is not key this entire paragraph can be ignored but for completeness) At the moment my only question is about the Monte Carlo sampling. Supplementary Table 2 of Goodwin et al. 2018b shows 18 parameters. With $3 \times 10^6$ members you're effectively taking a bit over 5 steps in each parameter axis ($18^5 \approx 2 \times 10^6$). This appears to be a fairly sparse sampling, which could be a problem no? I wasn't convinced by Goodwin et al. 2018b, 'This observation-consistent ensemble displays good agreement with the full ranges for all the observational quantities (Supplementary Table 4), which demonstrates that the $3 \times 10^4$ simulations have a good coverage of observational parameter space.' It seems perfectly plausible to me that the 95% ranges could agree but the distributions themselves are otherwise very different. If you've considered this before and can include the answers in the paper or point to them in the paper that would be great, if not then a sentence highlighting this and saying that they're areas for future research would suffice."*

**Authors' response: For completeness (with no bearing on the manuscript as method is not tied to the WASP model).** There is no contradiction in the manuscript.

The generation of the probabilistic ensemble involves an initial ensemble of $10^8$ simulations based on input parameter distributions that reflect prior knowledge of Earth system properties (such as climate sensitivity). These input parameter distributions are independent of one another (i.e. we do not assume that we know how one input parameter affects the value of other input parameters). From this initial ensemble of $10^8$ simulations, $3 \times 10^6$ are extracted to form the final probabilistic ensemble using an observational consistency test.

Some Monte Carlo sampling methods (effectively) ascribe a graduated weighting to each simulation according to the simulated position on the probability distribution of each of the observables used to constrain the system. For example if each simulated value agreed with the best estimate of each observable then that simulation would have the highest weighting. Here, instead of a graduated weighting, the weighting for each ensemble member is either 1 (included as observation-consistent) or 0 (excluded as inconsistent), based on whether the simulation agrees with the observable quantities to 95% confidence (see Goodwin et al, 2018b for a full methodological description including how the tails of the distribution are included).

Primarily, this accept/reject approach is taken because we assume we are able to ascribe some confidence range for each historical observational reconstruction, but we do not assume to know the shape of the probability distribution. To see why, consider the historic global ocean heat content anomaly. The observational consistency test in WASP for historic whole ocean heat content anomaly is based on 6 published reconstructions (e.g. see Goodwin et al., 2018b – see fig. 1 panels c and d therein for the six different reconstructions of ocean heat content anomaly considered). These reconstructions range from a best estimate of the increase in whole-ocean heat content from years 1971 to 2010 from under +200 ZJ to over +300 ZJ.

Considering this variation between the six reconstructions, a significant component of uncertainty in historic global ocean heat content anomaly is due to systematic uncertainty in the methodology used to convert sparse observations into a global mean, and

not random uncertainty in the underlying sparse observations themselves. Producing a probability distribution for ocean heat content anomaly would thus entail a highly subjective judgement on the relative merits of each different methodology used by the six different observational reconstructions. To avoid this highly subjective judgement (required for a graduated weighting approach), we use an accept/reject approach, based on whether a simulation lies between the minimum to maximum values of the 95% uncertainty ranges for each of the six reconstructions. (i.e. a simulation is 'observation-consistent' with whole ocean heat content anomaly reconstructions if it lies within the 95% uncertainty ranges of any of the six reconstructions). Therefore, the 95% probability range is known, but the shape of the probability distribution is unknown.

Also, in terms of numbers of simulations in the ensemble it should be noted we have generated $10^8$ simulations, from which we have extracted $3 \times 10^6$ simulations. Therefore, our initial sampling is adequate.

---

## Author Response (AR1)

We thank the reviewers and editor for their comments, which have greatly improved this revised version. Below, we spell out how we have amended the manuscript to each specific comment. The 'tracked changes' version of the manuscript follows this.

**Authors' response to the editorial comment:**

The comment by Astrid Kerkweg drew attention to an editorial requirement:

*"In particular, please note that for your paper, the following requirement has not been met in the Discussions paper:*

*• "The main paper must give the model name and version number (or other unique identifier) in the title."*

*Please add the names/acronyms (WASP/LGRTC) of the models used/developed and their version numbers to the title upon your revised submission to GMD. Yours,*

*Astrid Kerkweg"*

Thank you for drawing this editorial requirement to our attention. We have now included the model version described in the paper (WASP-LGRTC-1.0) within the new title:

"A computationally efficient method for probabilistic local warming projections constrained by history matching and pattern scaling, demonstrated by WASP-LGRTC-1.0"

**Authors' responses to reviewer 1's comments:**

We thank reviewer 1, Dr. Christopher Smith, for important and insightful comments about our manuscript. Below we explain how we will use these comments to further improve our manuscript.

*"General comments*
*This paper describes a simple methodology for translating global mean surface temperature diagnostic output from a simple climate model (WASP, but in theory any model like MAGICC, FAIR, Hector could theoretically be used) into regional surface temperature changes using a pattern scaling approach. While this is not a necessarily new concept (see fldgen: https://www.geosci-model-dev.net/12/1477/2019/), it is appreciated that a quick and simple tool would be greatly useful for translating the output of simple climate models (e.g. from IAMs) to regional impacts. Additionally, there is a nice link from carbon emissions/carbon budgets to future carbon emissions. With this knowledge it could be possible to assess regional impacts as a function of the remaining carbon budget (e.g. to 1.5C)."*

**Point 1: Applicability of the spatial tool to any model capable of projecting global mean warming.** We agree that any model capable of generating probabilistic projections for global mean surface temperature could be combined with the spatial tools presented in this manuscript. This would then generate projections for the mean and standard deviation for future local warming. In our revised manuscript we highlight how the spatial tool can be coupled to any model projecting global mean warming, and not just the WASP model. This is first explicitly stated in the revised title of the manuscript, referring to a 'method' as opposed to a 'model': "A computationally efficient method for probabilistic local warming projections constrained by history matching and pattern scaling, demonstrated by WASP-LGRTC-1.0". We then exolicitly state that we are presenting a method that can be coupled to any efficient model in the manuscript, e.g. page 1, Lines 17-20:

"This study presents a computationally efficient method for generating probabilistic projections of local warming across the globe, using a pattern scaling approach derived from the Climate Model Intercomparison Project phase 5 (CMIP5) ensemble, that can be coupled to any efficient model ensemble simulation of global mean surface warming."

Page 3, Lines 15-16:

"In this study, we present a new method for combining the LGRTC approach with an arbitrary efficient Earth system model to generate computationally efficient local warming projections for arbitrary forcing scenarios."

This change makes the revised manuscript more general in nature, and of interest to a wider range of readers.

**Point 2: Link to emissions budgets and regional impacts.** We agree that the approximation tool presented provides a useful link to assess regional impacts from carbon emissions/carbon budgets. In our revised manuscript we have improved this link by exploring the spatial warming pattern (the LGRTC) for the RCP2.6 scenario (Fig. 1a,d) – a scenario with strong mitigation and a high likelihood of meeting the Paris Climate Agreement goals.

*"p4 l4-10: I am not sure if three scenarios that all show various rates of continually increasing warming are sufficient to make this conclusion. I would suspect that this does not hold for RCP2.6 where most models stabilise in temperature but regional patterns may continue to evolve. It would be good to show this. It would be helpful to see the 1pctCO2 scenarios for comparison in figure 1, also. (also relevant to p6 l6-10)"*

**Scenarios with increasing warming and scenarios with stabilised warming.** We agree that the scenarios considered have increasing warming {although we note that in RCP4.5 the there is little additional warming after 2080 across the 13 CMIP5 models considered by Goodwin et al., (2018b – see figure 2c therein, grey shaded area), we agree that there is little time for the warming pattern to continue to evolve}.

We also agree that the RCP2.6 scenario offers a chance to explore our LGRTC tool for generating local warming projections in a scenario with stabilised climate. In this revised version we produce a LGRTC for RCP2.6 (Fig. 1a,d) and compare this to the existing scenarios (Table 1). This allows us to test our methodology for a climate with warming stabilised close to the Paris Climate Agreement warming targets of 1.5 and 2.0 °C.

*"p4l22: a point on different non-CO2 forcers in the three scenarios - the RCPs are quite heterogeneous in their aerosol forcing in future scenarios, and 1pctCO2 does not include them. I'm not sure this gives us much information for pattern scaling for custom emissions scenarios. See figure 3 in Liu et al. for temperature responses to - admittedly somewhat extreme - cases of aerosol forcing in Europe and Asia. https://doi.org/10.1175/JCLI-D-17-0439.s1 . Some more discussion about how this model could handle widely varying timeseries of global and regional aerosol forcing would really help strengthen the model (and paper)."*

**Pattern scaling for custom emissions scenarios with extreme aerosol emissions.** Our analysis now demonstrates how a single LGRTC with small uncertainty covers both RCP4.5 and RCP8.5 scenarios (Fig. 2c,f,i), and so will also be applicable to similar scenarios. In the production of the RCP scenarios, assumptions have been made as to the relative amounts of different forcing agents emitted (e.g. $CO_2$, other well mixed greenhouse gasses, aerosols etc) (e.g. see IPCC, 2013 or Meinshausen et al., 2011). Some forcing agents, specifically aerosols, are not well mixed in the

atmosphere but exert a significant local influence. We agree that for extreme cases of localised aerosol radiative forcing the pattern of warming would be different to the assumptions used in the generation of the different RCP scenarios.

In a revised manuscript, we state how our generic LGRTC approach can be applied to scenarios with similar underlying assumptions to the RCP scenarios (Page 6, Lines 33-35):

"The *generic ≥2°C* LGRTC pattern (Fig. 2) assumes idealised future pathways within the range of the RCP4.5 and RCP8.5 scenarios (Figure 3b,c), including a similar ratio of $CO_2$ to non-CO2 radiative forcing and spatial emissions of anthropogenic aerosols."

We then also explain how our generic LGRTC approach cannot be applied to scenarios with extreme spatial aerosol forcing that differs widely from the RCOP scenarios (p. 6 Lines 35-37):

"This *generic ≥2°C* LGRTC field should not be used for extreme scenarios that differ widely from the underlying societal assumptions of the RCP sceanrios, for example in their spatial aerosol forcing (e.g. see Liu et al., 2018)."

The method presented could potentially be extended in future work to attempt to calculate the impact on the LGRTC of different localised aerosol forcing patterns. For example by:
(1) Calculating the LGRTC for a scenario with $CO_2$ only (or well mixed greenhouse gas only) forcing, (2) Exploring the sensitivity of LGRTC patterns to regional aerosol emission in complex climate models, for cases with idealised aerosol emissions for each region (e.g. Europe, Asia etc).
(3) Combining the LGRTC from well mixed greenhouse forcing with the LGRTC from idealised aerosol forcing in each region, using knowledge of the relative emissions from greenhouse gasses and aerosols by region in the custom scenario, to generate a custom warming pattern for a scenario with extreme aerosol emission patterns. The key to such a method working is that the uncertainties introduced by the assumptions involved in combining different spatial patterns are smaller than the uncertainties introduced by the range of CMIP class model responses for a given scenario.

Since this potential method relies on new targeted experiments with complex CMIP models, this is not feasible to conduct for this paper and is reserved for future study.

Note that the calculations for the LGRTC presented here will apply for custom scenarios that do utilise similar assumptions to the RCP scenarios in terms of the relative amounts of well mixed greenhouse emissions and aerosols.

**Minor/stylistic points.** We also thank Dr. Smith for also raising minor/stylistic points, which we can confirm have all be addressed in the revised version.

**Reply to reviewer 2's comments**

*"1 General comments*
*Goodwin et al. present a tool for projecting local warming with uncertainty from multiple anthropogenic emissions scenarios. The major advance of the paper is the combination of output from a probabilistic climate model and warming ratios from AOGCM/ESMs (I note that the MAGICC/SCENGEN, http://www.cgd.ucar.edu/cas/wigley/magicc/, tool does a similar thing but given that this paper is not tightly coupled to MAGICC or any other probabilistic climate model and its code is open sourced I consider this paper to be a significant advance on the MAGICC/SCENGEN tool). I feel that this advance could be a very useful addition to the literature if a few concerns were addressed to provide more confidence in the paper's conclusions."*

**General comments.** We are pleased the reviewer sees the advance offered, and in this revision we have amended the manuscript to address the specific concerns raised – please see details below.

*"My major concerns focus on: whether the tool is actually scenario specific or not, how uncertainties from the climate model and LGRTC are combined and and whether WASP is actually a key part of the tool or whether any probabilistic climate model could be used.*

*One other key comment, given the availability of CMIP6 model output, I feel this paper could be significantly improved if it were to use CMIP6 output rather than focussing on CMIP5."*

**Concerns.** We spell out in detail below how our revision addresses the concerns raised. In brief, the revised manuscript:
(1) Analyses the LGRTC for an additional scenario, RCP2.6 (new Fig. 1a,d), and provides more robust statistical comparisons of the differences in LGRTC for the different scenarios (new Table 1; new Fig. 2), including identifying a spatial domain over which more generic LGRTC fields can be applied (Fig. 2a,b,c);
(2) Stresses that the method is not specific to the WASP model, and makes it clear that our methodology can be applied to any efficient model generating projections of global mean surface warming (Changed title; plus e.g. Page 1 Lines 17-20); and
(3) Reserves the analysis of CMIP6 model output for future study.

*"2.1 Scenario specificity of pattern scaling*

*It is not clear to me that the pattern scaling technique here is actually scenario agnostic. All the presented results are scenario specific (the RCP45 projections use RCP45 LGRTC and the RCP85 projections use RCP85 LGRTC) and there is no analysis of whether a 'general LGRTC' can be used nor whether such a 'general LGRTC' would have small enough uncertainties as to be useful."*

**Scenario specificity of the LGRTC.** We agree that the LGRTC pattern scaling technique is not truly scenario agnostic – there are of course factors about a scenario that affect the LGRTC. However, this revised version explores, by comparing the LGRTC for three RCP scenarios, how the LGRTC approach can be applied over scenarios that are similar to, yet not precisely the same as, the specific RCP scenarios.

*"I feel the comment (page 6, line 10), 'This allows future users to choose the spatial pattern scaling that is most suitable for their scenario.' is misleading. Only 3 patterns are available and none of them have been shown to be applicable for an emissions scenario different to the one from which they were derived (see comment above). Such cross-validation would be a vital step to providing confidence that the spatial pattern derived from one scenario can then be applied to any arbitrary scenario."*

Agreed that the sentence "This allows future users to choose the spatial pattern scaling that is most suitable for their scenario' was unclear, and this statement has been removed in the revised version. We now quantify how different the scenario LGRTC fields are, in terms of the ratio of inter-model variation to inter-scenario variation in the LGRTC (Table 1). We also combine the scenarios

Note that the arbitrary and generic ≤2°C LGRTC patterns (that include RCP2.6 information) are not practical to use because of the large uncertainties in the LGRTC, caused by the large inter-model differences in the RCP2.6 LGRTC patterns for CMIP5 models (Fig. 1d). However, for most of the globe the variation between CMIP5 model LGRTC patterns is still larger than the variation between scenarios (Fig. 2a,b – consider the regions with a vlid domain). Therefore, the approach is valid for a large domain, it just results in high uncertainty.

See below for details on broader points.

*"I am not convinced by the comment (page 4, line 8), 'The absolute value of differences in LGRTC between the three scenarios was below 0.72C perC in all grid-cells and mostly below 0.2C perC over the continents. Therefore, the choice of the emission scenario to define spatial pattern of warming in this study is not much relevant when only inhabited regions are considered.' Relative to strong mitigation targets (e.g. the 1.5C target), I am not convinced these are trivial variations. In addition, in this context 'mostly' is meaningless and provides no quantification of how wide the disagreement is nor of the regions in which this generalisation doesn't hold (and how wrong it is)."*

*"I am also not convinced by the comment (page 4, line 19), 'This might have led to the large differences in the Arctic region, but detailed analysis and explanation is outside the scope of this study.' If the pattern scaling approach is to be used for arbitrary scenarios, there needs to be evidence that one pattern, with sufficiently large uncertainties, can be applied to multiple scenarios and give results that are in line with known results from CMIP models. Any differences need to be explained as they are of key interest when applying this tool (or the tools' domain of applicability should only be limited to those regions where the differences are small/well understood)."*

**Imprecise wording of comparisons between scenario-LGRTC patters.** Agreed that the highlighted sentences do not provide robust statistical analysis of the differences and similarities between the LGRTC patterns for the scenarios. In a revised manuscript, we now provide a robust statistical comparison of the LGRTC for different scenarios (Table 1) and over different areas of the domain (Figure 2g,h,i). We now define the domain over which the tool is applicable for more generic scenarios that are similar to, but not identical to, the RCP2.6, RCP4.5 and RCP8.5 scenarios. See answer to the next paragraph for more details.

*"I think the data is there to address this concern. One suggestion (which would satisfy me) would be to derive some 'general LGRTC' (including uncertainty) which could be used with any emissions scenario. The 'general LGRTC' could then be applied to the RCPs (here meaning all RCPs, including RCP26 and RCP60, not just RCP45 and RCP85) and the differences quantified. This would provide a meaningful quantification of how big the uncertainties need to be on a 'general LGRTC' for it to sufficiently capture the variation across CMIP models and scenarios in the cases where we have data. I would be even more convinced if a 'general LGRTC' derived from CMIP5 RCPs was shown to hold for CMIP6 SSP scenarios."*

**Addressing concern over applicability of LGRTC for different scenarios/scenario dependence quantification.** Agreed that greater insight into the amount of scenario-dependence is required, and that this affects the validity of using the LGRTC for scenarios other than the scenarios from which they were derived. The following changes improve the manuscript:

(1) The revised manuscript analyses the LGRTC for an additional scenario, RCP2.6 (a stabilisation scenario with strong mitigation in line with the Paris Climate Agreement's targets of keeping warming under 2.0 °C): Fig. 1a,d). This scenario shows greater inter-model variation (Fig.1d comnpared to Fig 1e,f). In part, this greater model variation in LGRTC will likely be due the lower global mean warming in the RCP2.6 scenario: since the LGRTC has the global mean warming on the demoninator, scenarios that have small global mean warming will likely show more variation in spatial LGRTC patterns for different models.

(2) The revised manuscript provides a meaningful statistical comparison of the LGRTC for the different scenarios (RCP8.5, RCP4.5 and RCP2.6: Table 1). This statistical comparison constitutes a comparison of the magnitudes of LGRTC uncertainty due to differences 'within scenario but between CMIP5 models' to the LGRTC differences 'between scenarios'. i.e. comparing sigma_LGRTC within a scenario to the differences between mu_LGRTC for different scenarios.

(3) The revised manuiscript explores the feasibility of defining a domain over which a set of generic LGRTC patterns can be defined (with uncertainties large enough to capture variation across CMIP5 models and variation between scenarios).

Three generic LGRTC patterns are produced (Fig. 2): (i) an arbitrary scenario for any warming level (made by combining RCP2.6, RCP4.5 and RCP8.5); (ii) a generic scenario for warming up to 2 °C (mode by combining RCP2.6 and RCP4.5); and (iii) a generic scenario for wearming of 2 °C and more (mode by combining RCP2.6 and RCP8.5.

The methods used to produce the generic LGRTC fields are explained in Section 3.2.1, including equations (2) and (3). Ultimately, a key property of a generic LGRTC pattern must be that the uncertainty introduced by the scenario choice is less than the uncertainty introduced by the range of CMIP-class model responses within each a given scenario. Therefore, the revised manuscript restricts the domains of the generic LGRTC patterns to locations where the condition $|\mu_j - \mu_k|/\sigma_{LGRTC} < 1.0$ is met. This comparison also quantifies the value of $|\mu_j - \mu_k|/\sigma_{LGRTC}$ over the globe (Fig. 2g,h,i).

We will reserve comparisons to CMIP6 for future study.

*"2.2 Scenario specificity of WASP*

*WASP currently requires exogenous estimates of non-CO2 radiative forcing (see manuscript paragraph starting page 7, line 33). As far as I can tell, this means that this tool is not applicable to arbitray emissions scenarios but rather only ones for which there is an available non-CO2 radiative forcing quantification. I feel this is a rather fatal flaw of a tool which is meant to be applicable to arbitrary emissions scenarios.*

*An easy remedy would be to alter the tool from being 'WASP/LGRTC' to 'a general framework for coupling probabilistic climate model output and LGRTC' (insert acronym here) i.e. remove the explicit dependence on WASP. I can't see any reason why WASP is the only model with which this tool would work. This paper could still illustrate the use of the framework with WASP output, but such a reframing would make clear that the coupling could be done with any probabilistic climate model so a model which can run fully GHG-emissions driven could be used instead and would immediately fix the issue of WASP's limited available scenario set."*

Agreed, the method's (non)reliance on WASP is now explicitly stated. We agree that the LGRTC method can be applied to any arbitrary probabilistic climate model ensemble, not just the WASP ensemble used in the study. We have reframed the manuscript title and text in terms of offering a general framework, with WASP the efficient model used as an example tool to illustrate the approach.

*"2.3 Combination of uncertainties*

*I am not convinced that the combination of uncertainties in equation 2 is correct. In equation 2, shouldn't the resulting distribution be the product/convolution of the two distributions rather than the output of random sampling from the two distributions? Given LGRTC is assumed to be gaussian, and that the WASP output is approximately gaussian, wouldn't it be better to derive the distribution of Delta T_i (x, y, t) by taking the product of two gaussians (see e.g. https://ccrma.stanford.edu/~jos/sasp/Product_Two_Gaussian_PDFs.html) which isn't the same as the product of two gaussian variables (see e.g. https://math.stackexchange.com/questions/101062/is-the-product-of-twogaussian-random-variables-also-a-gaussian). I'm happy to be corrected on this as I am not a statistical expert. However, regardless of whether I am correct or not I think some explanation must be added to the manuscript or the supplementary to explain this uncertainty propogation."*

**Combination of uncertainties.** We agree with the statistical points made about the random sampling of two Gaussian distributions not in general giving the same answer as the convolution of two Gaussian distributions.

However, to make our LGRTC method applicable to any arbitrary probabilistic projection of global mean surface warming (not just from this WASP ensemble), we cannot assume that the projection of global mean surface warming is Gaussian. It may be that a projection of global mean surface warming is significantly skewed, for example due to a skewed probability distributiuon of climate sensitivity with a long tail of high values (e.g. see Assessment Report 5 of the IPCC, 2013). Therefore, to ensure that our approach is generally applicable to any efficient model's projection of global mean surface warming, we cannot take product of two Gaussian distributions as suggested by the reviewer. We explain this on Page 7 Lines 9-13:

"Note that eq. (2) does not assume that the distribution of global mean temperature projections, $\bar{\Delta}(T\_i)(t)$, from the efficient Earth system model is Gaussian. The distribution of $\bar{\Delta}(T\_i)(t)$ may not be Gaussian if, for example, the assumed climate sensitivity distribution has a long tail of high values (e.g. see Knutti et al., 2017). Thus, this method for generating the local warming distribution, eq. (2), can be applied to any arbitrary distribution of global mean surface warming from any arbitrary efficient climate model."

Our method, of randomly sampling from both the distributions of global mean warming and LGRTC, is applicable to any arbitrary projection of global mean surface warming from any arbitrary efficient Earth system model.

We now also point out that in the MATLAB approximation tool, which does tie in to the WASP ensemble, we have used the product of two Gaussian distributions (as the reviewer suggests) rather than random sampling (Page 8 Lines 26-27):

"Note that in this approximation tool the uncertainty in local warming is calculated directly by multiplying the assumed Gaussian distributions of LGRTC and global mean warming, eq. (8)."

*2.4 Reliance on WASP*
*It is not clear if this paper is using an existing WASP probabilistic distribution or presenting a new one (e.g. contradiction between page 5, line 9: '3x10[6] members' and page 2, line 23: '10[8] simulations'). If the reframing suggested earlier were to take place then this is no longer an issue (as the choice of particular probabilistic climate model is just for illustration and isn't a key feature of the tool). However, if this particular WASP probabilistic distribution is key then I would have to consider that component more closely."*

**The (non)reliance on WASP.** The novel methodology (of combining the LGRTC with a probabilistic ensemble of global mean warming from an efficient numerical model) is not tied to WASP. Therefore, we will be making the reframing suggested by the reviewer earlier clear in a revised manuscript. We adopt the probabilistic ensemble generated in Goodwin et al (2018b). We will make the particular ensemble used clear in the revised manuscript.

We note that there is not a contradiction between having 3x10[6] members in the posterior ensemble and 10[8] members of the prior ensemble in the WASP methodology (see below).

*"(If the WASP probabilistic distribution is not key this entire paragraph can be ignored but for completeness) At the moment …*

Our revised manuscript now presents a LGRTC method that can be applied to any efficient model's projection of global mean surface warming (rather than specific to only the WASP model). Therefore, the WASP probability distribution is not key to the manuscript's findings, and so the points made by the reviewer in this paragraph are not relevant – as the reviewer identified.

[revised manuscript text omitted]

(a) $\mu_{LGRTC}$ (RCP2.6 peak)  (b) $\mu_{LGCRT}$ (RCP4.5)  (c) $\mu_{LGRTC}$ (RCP8.5)  (°C per °C)

(d) $\sigma_{LGRTC}$ (RCP2.6 peak)  (e) $\sigma_{LGRTC}$ (RCP4.5)  (f) $\sigma_{LGRTC}$ (RCP8.5)

**Figure 1: The LGRTC in RCP2.6, RCP4.5 and RCP8.5 scenarios analysed from a multi-model ensemble of CMIP5 simulations.** **(a), (b) and (c) show the multi-model mean LGRTC, $\mu_{LGRTC}$, while (d), (e) and (f) show the multi-model standard deviation in LGRTC, $\mu_{LGRTC}$, for each scenario.**

[Figure]

**Figure 2: The LGRTC in the *arbitrary, generic ≤2°C* and *generic ≥2°C* scenarios. Panels (a), (b) and (c) show the scenario mean LGRTC. Panels (c), (d) and (e) show the scenario standard deviation in LGRTC. Panels (g), (h) and (i) show the ratio of the maximum absolute discrepancy in the mean LGRTC from the underlying RCP scenarios, $\Delta\mu$, to the standard deviation in the LGRTC, $\sigma$, in the combined scenario: $\Delta\mu/\sigma$.**

[Figure]

**Figure 3: Projections of global mean surface warming from the history matched WASP ensemble for different future carbon emission sizes. (a) Frequency distributions of projected warming in the WASP ensemble for different future carbon emission sizes after the start of 2018. (b) Ensemble-mean global warming as future cumulative carbon emitted increases. (c) Ensemble standard**
5   **deviation in global warming as future carbon emitted increases. (b) and (c) show the RCP8.5 (blue), RCP6.0 (red), RCP4.5 (orange) and RCP2.6 (purple) scenarios. A quadratic approximation, eq. 3 for (b) and eq. 4 for (c), is a good fit to the RCP8.5 scenario (thin black line). All panels show warming calculated relative to the 1850-1900 average.**

[Figure]

**Figure 4: Projected warming for the period 2081-2100 relative to the 1850-1900 average from $1 \times 10^3$ history matched simulations of the ultra-fast WASP/LGRTC ensemble. The left-hand column is for the RCP4.5 scenario and the right-hand column is for the RCP8.5 scenario. The top, middle and bottom rows represent the mean, 83rd percentile and 17th percentile of the model ensemble.**

[Figure]

(a) Central warming projection, future carbon emission = 500PgC

(b) Probabilistic warming projection, future carbon emission = 500PgC

**Figure 5: Warming projections when future emissions reach 500 PgC from the start of 2018. (a) The spatial distribution of the central warming projection. (b) The probability distributions of local warming for 7 locations (solid colour lines) and the global surface average (black dashed line). All warming projections given relative to the average temperature from 1850 to 1900. Global mean warming projected from the quadratic approximation to the history matched WASP ensemble (eqns. 3 to 6) using the generic ≥2°C spatial pattern.**

---

## Referee Report (RR1)

**General comments**

As discussed previously, Goodwin et al. present a tool for projecting local warming with uncertainty from multiple anthropogenic emissions scenarios. The major advance of the paper is the combination of output from a probabilistic climate model and warming ratios from AOGCM/ESMs.

The revisions provided have been very helpful. My remaining major concern focuses on clarifying the tool's domain of applicability.

**Major concerns**

**Applying the tool to $<=$ 2C scenarios**

The discussion of the issues with applying the tool for $<=$2C scenarios, particularly the LGRTC maps for $<=$2C scenarios is buried far too deep into the paper. The revised line which raises this issue is, "The arbitrary and generic $<=2°$C LGRTC scenarios are problematic to use in practice" (page 6, line 23 of diff pdf). I don't think this is a serious enough problem to prevent publication of the paper. However, I do think that it is a serious enough coveat of this work to warrant discussion in the abstract. At the moment, without a close reading, the paper gives the impression that the tool can be used for arbitrary warming and cumulative emissions targets but, as acknowledged by the authors, that is not the case. For example, a line like, "While the tool can assess arbitrary scenarios, using it for scenarios with peak warming $<=2°$C is problematic due to the large uncertainties involved."

Given this, I don't think it is appropriate for the headline figure (which I would argue is Figure 5) to use a scenario which is only just above the 2C level (given the issues acknowledged by the authors about using their tool for warming in the region of 2C). I would recommend that the headline Figure 5 use a scenario with warming of at least 2.5C, to avoid straying into ambiguous territory. I also note that in their provided matlab code the authors use a threshold of 1.95C to decide whether to use their "generic $>=$ 2C" map or not, which seems an odd choice given their paper explicitly says don't use the $>$2C pattern for targets less than 2C.

**Calculating LGRTC for RCP2.6**

It is of some concern that the authors have to use a different method to calculate the LGRTC for RCP2.6 (page 4, line 10-13 of diff pdf). I think that it also raises questions about the statement (page 3, line 41-42 of diff pdf), "To first order, the mean LGRTC can be treated as being independent of time and emission scenarios (Leduc et al, 2016, 2015).", it appears here as if the authors have shown that doesn't hold for strong mitigation scenarios? I understand that the need for a new method arises because a 2006-2025 reference period is used when calculating the LGRTC. However, given the authors' focus on warming relative to 1850-1900, I don't completely understand the logic of calculating

LGRTC with a 2006-2025 reference period in the first place (and I can't find any justification for doing so in the manuscript, please correct me if I've missed something). Having to use a different method for strong mitigation scenarios is a troubling sign and I suspect it is a large part of the reason that their "generic $<=$ 2C" LGRTC is not useable in practice.

**Addressing the concerns**

I think both my major concerns can be addressed by simply being clearer about the tool's limited domain of applicability throughout the manuscript, particularly in the key abstract, introduction and conclusion sections. Providing a tool for higher warming levels is nonetheless a useful contribution, and I would not object to work on a tool for lower warming levels being left for future work.

**Minor concerns**

**Reproducibility**

I note that the authors have included much of the code required to produce their paper and commend them for their efforts to do so. I am torn because a key piece is missing, however I know how difficult that piece is and can sympathise with why it hasn't been included. The missing pieces is the code required to derive the LGRTC patterns in the first place. Nowhere is such code provided. In practice, I know that deriving these patterns is generally difficult and requires all sorts of programming gymnastics. If it is possible to provide, I think that would be great and would complete the authors' existing reproducibility efforts.

---

## Author Response (AR2)

**Reply to reviewer comments:**

We thank the editors and reviewers for their careful reading of the manuscript and helpful comments. Below we detail how we have amended both the manuscript and model code to address the specific comments made.

*Reviewer comments:*

*General comments*

*As discussed previously, Goodwin et al. present a tool for projecting local warming with uncertainty from multiple*
*anthropogenic emissions scenarios. The major advance of the paper is the combination of output from a probabilistic climate model and warming ratios from AOGCM/ESMs.*

*The revisions provided have been very helpful. My remaining major concern focuses on clarifying the tool's domain of applicability.*

We agree that the major advance of this paper is the combination of output from a probabilistic climate model and warming ratios from AOGCM/ESMs. We also agree that, while the domain of applicability had been stated, it had not been stated with enough clarity in the previous version of the manuscript. Below we detail how we have clarified the domain of applicability in the abstract, manuscript and released model code in response to your detailed points.

*Major concerns*
*Applying the tool to <= 2C scenarios*
*The discussion of the issues with applying the tool for <=2C scenarios, particularly*
*the LGRTC maps for <=2C scenarios is buried far too deep into the paper.*
*The revised line which raises this issue is, "The arbitrary and generic <=2°C*
*LGRTC scenarios are problematic to use in practice" (page 6, line 23 of diff*
*pdf). I don't think this is a serious enough problem to prevent publication of*
*the paper. However, I do think that it is a serious enough coveat of this work to*
*warrant discussion in the abstract. At the moment, without a close reading, the*
*paper gives the impression that the tool can be used for arbitrary warming and*
*cumulative emissions targets but, as acknowledged by the authors, that is not*
*the case. For example, a line like, "While the tool can assess arbitrary scenarios,*
*using it for scenarios with peak warming <=2°C is problematic due to the large*
*uncertainties involved."*

We agree that this caveat needs to be in the abstract to achieve clarity for the reader. We now include a line in the abstract as suggested (Line 20-21):

"While the method can project local warming for arbitrary future scenarios, using it for scenarios with peak global
mean warming $\leq 2°C$ is problematic due to the large uncertainties involved."

*Given this, I don't think it is appropriate for the headline figure (which I would*

*argue is Figure 5) to use a scenario which is only just above the 2C level (given*
*the issues acknowledged by the authors about using their tool for warming in*
*the region of 2C). I would recommend that the headline Figure 5 use a scenario*
*with warming of at least 2.5C, to avoid straying into ambiguous territory. I also*
*note that in their provided matlab code the authors use a threshold of 1.95C to*
*decide whether to use their "generic >= 2C" map or not, which seems an odd*
*choice given their paper explicitly says don't use the >2C pattern for targets*
*less than 2C.*

Agreed that the original figure 5 was plotted for a scenario that was close to (but above) 2 °C global mean surface warming, and that it is clearer to show the figure for warming for a scenario that is further above 2 °C. We have re-plotted this figure for future emissions of 600PgC, giving warming further above 2 °C (now 2.4 °C).

We have also amended the MATLAB code so that it only displays the figure only if the global mean surface warming is greater
than 2 °C. If the emission is set such that global mean surface warming is ≤ 2 °C, the MATLAB code now outputs: *'Global Mean Warming under 2 degrees C, increase emission size*.'

*Calculating LGRTC for RCP2.6*
*It is of some concern that the authors have to use a different method to calculate the LGRTC for RCP2.6 (page 4, line 10-13*
*of diff pdf). I think that it also raises questions about the statement (page 3, line 41-42 of diff pdf), "To first order, the mean*
*LGRTC can be treated as being independent of time and emission scenarios (Leduc et al, 2016, 2015).", it appears here as if*
*the authors have shown that doesn't hold for strong mitigation scenarios? I understand that the need for a new method arises*
*because a 2006-2025 reference period is used when calculating the LGRTC. However, given the authors' focus on warming*
*relative to 1850-1900, I don't completely understand the logic of calculating LGRTC with a 2006-2025 reference period in*
*the first place (and I can't find any justification for doing so in the manuscript, please correct me if I've missed something).*
*Having to use a different method for strong mitigation scenarios is a troubling sign and I suspect it is a large part of the reason*
*that their "generic <= 2C" LGRTC is not useable in practice.*

We agree that the statement on p3 line 42-42 did not appear consistent with the findings of the paper. Note that this was an issue with the statement in this paper and not the references in that statement. We have now clarified that this finding only refers to scenarios that do not reach peak warming before year 2100 (p3., lines 41-42):

"To first order, for scenarios that do not reach peak warming before 2100, the mean LGRTC can be treated as being
independent of time and emission scenarios (Leduc et al, 2016, 2015)."

We use the 2006-2025 period as the reference because we are using the simulated future local warming in each model to make our projections. Essentially, we are making a comparable assumption to IPCC Assessment Report 5 (Chapter 12) when the CMIP5 models are used to make projections of global mean temperature above the 1850-1900 average (see Collins et al.,
2013, Table 12.3 therein).

We agree that requiring a different method for mitigated scenarios (warming <= 2°C) is likely linked to the caveat that the method should only be applied to scenarios with warming >2 °C.

*Addressing the concerns*

*I think both my major concerns can be addressed by simply being clearer about the tool's limited domain of applicability throughout the manuscript, particularly in the key abstract, introduction and conclusion sections. Providing a tool for higher*

*warming levels is nonetheless a useful contribution, and I would not object to work on a tool for lower warming levels being left for future work.*

We agree that providing a tool for warming levels >= 2.0°C is a useful contribution. We now make the caveat clear in the abstract (as described above), and we have amended the MATLAB code so that it only makes projections of local warming if global mean warming is greater than 2.0 °C (see above).

*Minor concerns*

*Reproducibility*

*I note that the authors have included much of the code required to produce their paper and commend them for their efforts to*

*do so. I am torn because a key piece is missing, however I know how difficult that piece is and can sympathise with why it hasn't been included. The missing pieces is the code required to derive the LGRTC patterns in the first place. Nowhere is such code provided. In practice, I know that deriving these patterns is generally difficult and requires all sorts of programming gymnastics. If it is possible to provide, I think that would be great and would complete the authors' existing reproducibility efforts.*

Agreed that reproducibility is important. In this case, the reviewer is correct that the code for deriving the LGRTC patters is far more complicated than the code for running the models. Since the method of deriving LGRTC patterns was originally presented in Leduc et al (2015, 2016), we do not reproduce the code here. Readers can refer to those earlier studies for a full description of the numerical methods applied.

*Additional references:*

Collins, M., R. Knutti, J. Arblaster, J.-L. Dufresne, T. Fichefet, P. Friedlingstein, X. Gao, W.J. Gutowski, T. Johns, G. Krinner,

M. Shongwe, C. Tebaldi, A.J. Weaver and M. Wehner, 2013: Long-term Climate Change: Projections, Com- mitments and Irreversibility. In: *Climate Change 2013: The Physical Science Basis. Contribution of Working Group I to the Fifth Assessment Report of the Intergovernmental Panel on Climate Change* [Stocker, T.F., D. Qin, G.-K. Plattner, M. Tignor, S.K. Allen, J. Boschung, A. Nauels, Y. Xia, V. Bex and P.M. Midgley (eds.)]. Cambridge University Press, Cambridge, United Kingdom and New York, NY, USA.

[revised manuscript text omitted]